# PerFedMask: Personalized Federated Learning with Optimized Masking Vectors

**Mehdi Setayesh, Xiaoxiao Li, and Vincent W.S. Wong**[*]
Department of Electrical and Computer Engineering
The University of British Columbia
{setayeshm,xiaoxiao.li,vincentw}@ece.ubc.ca

## Abstract

Recently, various personalized federated learning (FL) algorithms have been proposed to tackle data heterogeneity. To mitigate device heterogeneity, a common approach is to use masking. In this paper, we first show that using random masking can lead to a bias in the obtained solution of the learning model. To this end, we propose a personalized FL algorithm with optimized masking vectors called PerFedMask. In particular, PerFedMask facilitates each device to obtain its optimized masking vector based on its computational capability before training. Fine-tuning is performed after training. PerFedMask is a generalization of a recently proposed personalized FL algorithm, FedBABU (Oh et al., 2022). PerFedMask can be combined with other FL algorithms including HeteroFL (Diao et al., 2021) and Split-Mix FL (Hong et al., 2022). Results based on CIFAR-10 and CIFAR-100 datasets show that the proposed PerFedMask algorithm provides a higher test accuracy after fine-tuning and lower average number of trainable parameters when compared with six existing state-of-the-art FL algorithms in the literature. The codes are available at `https://github.com/MehdiSet/PerFedMask`.

## 1 Introduction

Federated learning (FL) is a distributed artificial intelligence (AI) framework, which allows multiple edge devices to train a single model collaboratively (Konečnỳ et al., 2015; McMahan et al., 2017). The model is trained under the orchestration of a central server. In a typical FL algorithm, each communication round includes the following steps: (1) the edge devices download the latest model from the server to be used as their local model; (2) each device performs multiple local update iterations for updating the local model based on its local dataset; (3) the devices upload their updated local models to the server; (4) the server computes the new model by aggregating the local models. In practical systems, the devices may have diverse and limited computation, communication, and storage capabilities. Moreover, the local datasets available to the devices may be different in size, and contain non-independent and identically distributed (non-IID) data samples across the devices.

Under these heterogeneous settings, the performance of the conventional FL algorithms can degrade (Wang et al., 2020; Li et al., 2021). To handle the case when the data is non-IID, some works (Li et al., 2020a; Karimireddy et al., 2020) have introduced new optimization frameworks to obtain a more stable global model for the devices. Another approach to address the data heterogeneity issue is by designing a personalized model for each device (Arivazhagan et al., 2019; Fallah et al., 2020; Collins et al., 2021; Oh et al., 2022). In personalized FL algorithms, instead of obtaining a single model for all the devices, an initial model is obtained. This initial model can then be personalized for each device using its local data samples.

To overcome the computation limitation of the heterogeneous devices, one common approach is to use masking vectors. Masking vectors can be used to train only a sub-network of the learning model for each device based on the computational capability of that device. Masking vectors can be combined with pruning and freezing methods. Pruning methods utilize masking vectors to keep the important parameters of the learning model and remove those which are unimportant from

---

[*]Corresponding author: Vincent W.S. Wong

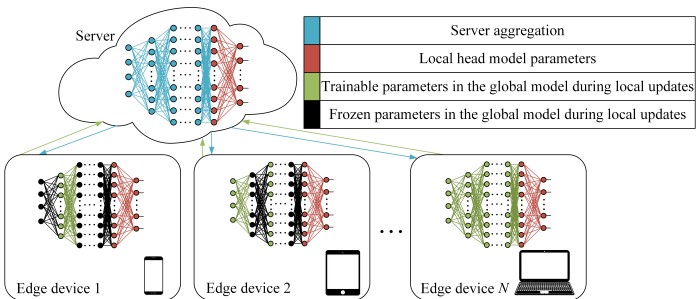

Figure 1: Illustration of an FL system using PerFedMask. The model is decoupled into a global model and a local head model. The local head model remains unchanged during training. The devices collaboratively train the global model. Some parts of the global model can be frozen for the devices during local updates using the optimized masking vectors. After training, a personalized model is obtained for each device by fine-tuning.

the model architecture. However, leveraging pruning in FL may incur additional communication overhead (Babakniya et al., 2022; Bibikar et al., 2022). Moreover, it results in different model architectures for the devices (Guo et al., 2016). This may lead to accuracy loss, particularly when data heterogeneity exists in the system (Hong et al., 2022). In the freezing methods, the masking vectors are used to freeze some parts of the learning model for each device. Unlike pruning, the masked parameters are not removed but are frozen during local updates. Hence, a more stable FL algorithm is obtained without changing the learning model architecture. Sidahmed et al. (2021) and Pfeiffer et al. (2022) have shown that freezing methods can reduce the computational and communication resources required for training the learning model in FL. However, the aforementioned works use heuristic approaches for designing the masking vectors, and do not provide a theoretical analysis for their choice. Also, the aforementioned works do not address the data heterogeneity issue in their proposed algorithms. In this work, we aim to answer the following question: *By exploiting freezing method in FL, what is a systematic approach to determine the masking vectors which can improve the final test accuracy in a setting with data and device heterogeneities?*

We first show that using the masking vectors to freeze the model parameters for the devices may lead to a bias in the convergence bound. This bias can hinder the success of employing masking vectors to tackle the device heterogeneity issue in FL. Using the insights from our analysis, we propose PerFedMask, **Per**sonalized **Fed**erated Learning with Optimized **Mask**ing Vectors (see Fig. 1). Specifically, by decoupling the learning model into a global model and a local head model, we first freeze the local head model for all the devices. Then, we freeze a portion of the global model for each device based on its computational capability. In our work, the masking vectors are determined before training through minimizing the bias term in the convergence bound. The bias can be mitigated by this approach. However, it may not be eliminated completely. Thus, after training of the global model, the frozen parameters of the local head model can assist to fine-tune the entire learning model for each device. We demonstrate empirically the effectiveness of PerFedMask under the heterogeneous settings when compared with six existing state-of-the-art FL algorithms.

PerFedMask has several distinct advantages: (1) PerFedMask generalizes the recently proposed personalized FL algorithm, FedBABU (Oh et al., 2022). In particular, FedBABU is a special case of PerFedMask when all the devices have the same computational capability. (2) PerFedMask is flexible. Since PerFedMask does not change the model architecture, it can be combined with other FL algorithms such as HeteroFL (Diao et al., 2021) and Split-Mix FL (Hong et al., 2022) to further improve the performance. (3) PerFedMask can address the objective inconsistency problem, which arises due to different number of local update iterations. Unlike FedNova (Wang et al., 2020) which requires the modification of device optimizers to tackle this issue, in PerFedMask, we consider the same number of local update iterations for all the devices, while adjusting the required number of computations for those devices with lower computational capabilities.

## 2 RELATED WORK

**FL Algorithms with non-IID Data.** FedAvg (McMahan et al., 2017) is the most popular FL algorithm, which aims to find a single model for all the devices. However, the local data samples at the

devices are usually non-IID, which can lead to data heterogeneity among the devices. To address this problem, Li et al. (2020a) proposed FedProx by introducing a proximal term in the objective function. However, this method does not attempt to find a personalized model for each device. To tackle this issue, different personalized FL algorithms have been proposed (Arivazhagan et al., 2019; Fallah et al., 2020; Collins et al., 2021; Oh et al., 2022) and most of them utilize fine-tuning to obtain a higher accuracy. In particular, after training, each device uses its local data samples to fine-tune the obtained model by performing several iterations of stochastic gradient descent (SGD). Personalization by fine-tuning may be limited if we focus on increasing the performance of a single model across all devices (Deng et al., 2020; Oh et al., 2022). Oh et al. (2022) proposed FedBABU, where after decoupling the learning model into a global model and a local head model, the same randomly initialized local head model is considered for all the devices. In FedBABU, the local head model is not updated during training. Nevertheless, the aforementioned methods are not designed specifically to tackle the device heterogeneity issue.

**FL Algorithms with Heterogeneous Devices.** Since different types of devices (e.g., smartphones, tablets, laptops) may participate in FL, the devices' capabilities vary from one another. Pruning method is one approach to address the device heterogeneity issue. Jiang et al. (2022); Babakniya et al. (2022); Bibikar et al. (2022) proposed different algorithms for consistent mask adjustment at the devices and mask aggregation at the server during training. Some works have considered static pruning at initialization (i.e., before training). Diao et al. (2021) proposed HeteroFL, where the local model parameters for each device are considered to be a subset of the learning model parameters. Hong et al. (2022) proposed Split-Mix FL, where a large model is split into several small base sub-networks according to the model width. Each device selects some of the base models for training. In general, the efficiency of different pruning techniques for obtaining an appropriate masking vector even in the centralized machine learning setting is still under investigation (Liu et al., 2022). Since the pruned connections are no longer available in the model architecture, pruning at initialization may inevitably degrade the performance of FL algorithms. Freezing method is another approach to address the device heterogeneity issue. Sidahmed et al. (2021); Chen et al. (2021); Pfeiffer et al. (2022) proposed different algorithms, where the masking vectors are used to freeze a portion of the learning model for each device in FL. Hence, while the required computation and communication resources can be reduced for each device, a more stable FL algorithm is obtained without changing the learning model architecture. In this paper, we apply freezing method in FL. We leverage the optimized masking vectors and fine-tuning to address both data and device heterogeneity issues. Appendix A contains a full discussion of the related work.

## 3  PROBLEM SETTING

We consider one server and $N$ edge devices. Each device $n \in [N] = \{1, 2, \ldots, N\}$ has its own set of local data samples $\mathcal{D}_n$. In a supervised learning setting, each device aims to find a learning model $\boldsymbol{\theta}_n \in \mathbb{R}^{d_{\boldsymbol{\theta}}}$ for predicting the true label $y_n$ given the input $\boldsymbol{x}_n$, $(\boldsymbol{x}_n, y_n) \in \mathcal{D}_n$, where $d_{\boldsymbol{\theta}}$ denotes the dimension of the learning model. Let $f_n(\boldsymbol{\theta}_n)$ represent the expected loss over the data distribution of device $n$. We have $f_n(\boldsymbol{\theta}_n) = \mathbb{E}_{(\boldsymbol{x}_n, y_n) \sim p_n} \mathcal{L}(\boldsymbol{\theta}_n; \boldsymbol{x}_n, y_n)$, where $\mathcal{L}(\boldsymbol{\theta}_n; \boldsymbol{x}_n, y_n)$ is the loss function that measures the prediction error of $\boldsymbol{\theta}_n$ over data samples $(\boldsymbol{x}_n, y_n) \in \mathcal{D}_n$, and $p_n$ is the distribution over $\mathcal{D}_n$. Formally, the optimization problem $\min_{\boldsymbol{\theta}} \frac{1}{N} \sum_{n=1}^{N} f_n(\boldsymbol{\theta})$ is solved in the conventional FL. All our results can easily be extended to the weighted averaging case, where the devices have different size of data samples. In this work, we study both data and device heterogeneity issues using model personalization and masking vectors.

**Model Personalization.** In the heterogeneous data setting, the probability distribution $p_n$ varies across the devices. Unlike conventional FL problems, to obtain a more personalized solution for each device, the learning models $\boldsymbol{\theta}_n$, $n \in [N]$, are not equal to each other. Similar to FedBABU (Oh et al., 2022), we decouple each learning model $\boldsymbol{\theta}_n$ into a global representation model $\boldsymbol{w}_g \in \mathbb{R}^{d_{\boldsymbol{w}}}$ and a device-specific head model $\boldsymbol{\phi}_n \in \mathbb{R}^{d_{\boldsymbol{\phi}}}$, where $d_{\boldsymbol{w}}$ and $d_{\boldsymbol{\phi}}$ denote the dimensions of the global model and local head model, respectively. We have $d_{\boldsymbol{w}} + d_{\boldsymbol{\phi}} = d_{\boldsymbol{\theta}}$. To further improve accuracy after training, we freeze the local head model parameters during training. In particular, before training, $\boldsymbol{\phi}_g$ is initialized randomly by the server to be used for initialization of all the devices' local head models. We have $\boldsymbol{\phi}_n = \boldsymbol{\phi}_g$, $n \in [N]$. $\boldsymbol{\phi}_n$ remains unchanged during training until the convergence is reached for the global model $\boldsymbol{w}_g$. The global model is obtained by minimizing the objective

function $F(\{\boldsymbol{w}_g, \boldsymbol{\phi}_g\}) = \frac{1}{N}\sum_{n=1}^{N} f_n(\{\boldsymbol{w}_g, \boldsymbol{\phi}_n\})$ throughout all the communication rounds, where the operator $\{.,.\}$ denotes the concatenation of the learning model parameters. Since $\boldsymbol{\phi}_g$ and $\boldsymbol{\phi}_n$ are never updated during training, with some abuse of notation, we consider the objective function as $F(\boldsymbol{w}_g) = \frac{1}{N}\sum_{n=1}^{N} f_n(\boldsymbol{w}_g)$. After convergence of $\boldsymbol{w}_g$, each device $n$ obtains its personalized local head model $\boldsymbol{\phi}_n$ by fine-tuning the learning model $\boldsymbol{\theta}_n$ using its local data samples.

**Masking Vectors.** In the heterogeneous device setting, devices vary in their computational and communication capabilities. We consider that in each communication round, the devices perform $\tau$ local update iterations. When deploying the same local models on all the devices, some devices with limited computational capability are not able to complete $\tau$ local update iterations and send their final local models to the server for aggregation in a timely manner. To address the device heterogeneity issue, masking vectors are used to freeze a part of the local model parameters by customization. A masking vector $\boldsymbol{m}_n \in \{0,1\}^{d_w}$ is selected for each device $n \in [N]$ based on its computational capability. During local update iterations, each device $n$ only updates those parameters in the global model that correspond to non-zero values of the masking vector $\boldsymbol{m}_n$. Other parameters, which correspond to the elements of $\boldsymbol{m}_n$ with zero values, are frozen during local updates. Note that in FedBABU (Oh et al., 2022), all the elements of vector $\boldsymbol{m}_n$ are equal to one for all devices $n \in [N]$.

Let $\boldsymbol{w}_n^i(t)$ denote the local model of device $n$ at the beginning of local update iteration $i$ in communication round $t$. At initialization (i.e., $i = 1$), we set $\boldsymbol{w}_n^1(t) = \boldsymbol{w}_g(t)$. At local update iteration $i > 1$, the local model of device $n$ is updated using SGD as follows:

$$\boldsymbol{w}_n^{i+1}(t) \leftarrow \boldsymbol{w}_n^i(t) - \eta(t)\boldsymbol{m}_n \odot \nabla f_n(\boldsymbol{w}_n^i(t), b_n^i(t)), \quad i = 1, \ldots, \tau, \tag{1}$$

where $\odot$ denotes the element-wise product, $\eta(t)$ is the learning rate, and $b_n^i(t)$ is the local batch sample chosen uniformly at random from the local dataset $\mathcal{D}_n$. After performing $\tau$ local update iterations, each device $n$ sends its final local model, i.e., $\boldsymbol{w}_n^{\tau+1}(t)$ to the server. We have

$$\boldsymbol{w}_n^{\tau+1}(t) = \boldsymbol{w}_g(t) - \eta(t)\boldsymbol{m}_n \odot \sum_{i=1}^{\tau} \nabla f_n(\boldsymbol{w}_n^i(t), b_n^i(t)). \tag{2}$$

Algorithm 2 in Appendix B describes the DeviceLocalUpdate function based on (1) and (2). In the aggregation step, we consider that the server aggregates the received final local models by taking the masking vectors of the devices into account. The global model for the next communication round can thus be determined through stable aggregation of unfrozen parameters, as follows:

$$\boldsymbol{w}_g(t+1) = \sum_{n \in \mathcal{N}(t)} \boldsymbol{k}_n \odot \boldsymbol{w}_n^{\tau+1}(t), \tag{3}$$

where $(\boldsymbol{k}_n)_l = \frac{(\boldsymbol{m}_n)_l}{\sum_{n' \in \mathcal{N}(t)} (\boldsymbol{m}_{n'})_l}$ denotes the $l$-th element of vector $\boldsymbol{k}_n$. $\mathcal{N}(t)$ denotes the set of devices participating in training in communication round $t$. Specifically, the server selects a fraction $c$ of $N$ devices at random as participating devices in each communication round. For each device $n$, vector $\boldsymbol{k}_n$ is obtained as a normalized masking vector using the vectors $\boldsymbol{m}_n, n \in \mathcal{N}(t)$. Using $\boldsymbol{k}_n$ in (3) indicates that the server only aggregates the updated parameters from the participating devices.

# 4 THEORETICAL RESULTS ON THE CONVERGENCE RATE OF THE GLOBAL MODEL

In this section, we analyze the convergence rate of the global model when masking vectors are used by the devices. Without loss of generality, we focus on non-convex loss functions. We also present the convergence results for strongly convex loss functions in Appendix C. In both cases, we assume that the loss functions are smooth. For simplicity, we obtain our convergence results for the full device participation scenario[1] (i.e., $c = 1$, $\mathcal{N}(t) = [N]$ for all $t$). The analysis relies on the following assumptions, which are commonly used for obtaining the convergence rate of different FL algorithms in the literature (Li et al., 2020b; Reddi et al., 2021; Amiri et al., 2022).

**Assumption 1.** *The function $f_n(\boldsymbol{w})$, $n \in [N]$, is L-smooth and satisfies:*

$$\left\| \nabla f_n(\boldsymbol{w}_n^i(t)) \right\|^2 \leq 2L \left( f_n(\boldsymbol{w}_n^i(t)) - f_n^* \right), \quad n \in [N], \quad i = 1, \ldots, \tau, \quad \forall t, \tag{4}$$

*where $f_n^*$ denotes the minimum value of $f_n(\boldsymbol{w})$.*

---

[1]Using the techniques presented in Li et al. (2020b); Karimireddy et al. (2020), the extension to the general case (i.e., $c \leq 1$) would be straightforward.

**Assumption 2.** $\nabla f_n(\boldsymbol{w}_n^i(t), b_n^i(t))$ *is an unbiased stochastic gradient of function $f_n(\boldsymbol{w})$. The variance of the masked stochastic gradients is bounded for each device $n \in [N]$. We have*

$$\mathbb{E}\big\|\boldsymbol{k}_n \odot \nabla f_n(\boldsymbol{w}_n^i(t), b_n^i(t)) - \boldsymbol{k}_n \odot \nabla f_n(\boldsymbol{w}_n^i(t))\big\|^2 \le \xi_n^2, \quad n \in [N], \quad i = 1, \ldots, \tau, \quad \forall t. \quad (5)$$

**Assumption 3.** *The expected squared $l_2$-norm of the masked stochastic gradients for all the devices is uniformly bounded. We have*

$$\mathbb{E}\big\|\boldsymbol{m}_n \odot \nabla f_n(\boldsymbol{w}_n^i(t), b_n^i(t))\big\|^2 \le G^2, \quad n \in [N], \quad i = 1, \ldots, \tau, \quad \forall t. \quad (6)$$

When the masking vectors are determined based on the computational capability of the devices, we define the term $\gamma_n = \max_l (\boldsymbol{k}_n)_l$ to quantify the degree of device heterogeneity in the network. Note that in the full device participation scenario, $\frac{1}{N} \le \gamma_n \le 1$, $n \in [N]$. In addition, $\gamma_n$ is inversely proportional to the minimum non-zero element of the vector $\sum_{n'=1}^{N} \boldsymbol{m}_n \odot \boldsymbol{m}_{n'}$. Hence, larger $\gamma_n$ implies a higher degree of device heterogeneity for device $n \in [N]$. We first present the following lemma, which is derived using Theorem 3 in Fang et al. (1994). From Lemma 1, we can quantify the impact of freezing the parameters by masking vectors on the convergence bound.

**Lemma 1.** *The following inequality holds for any vectors $\boldsymbol{x}$ and $\boldsymbol{z} \in \mathbb{R}^d$, for which there exists $Q > 0$ satisfying $\big|\min_l (\boldsymbol{x} \odot \boldsymbol{z})_l\big| \le Q$, and for any vector $\boldsymbol{y} \in \mathbb{R}^d$:*

$$\langle \boldsymbol{x}, \boldsymbol{y} \odot \boldsymbol{z} \rangle \le \max_l (\boldsymbol{y})_l \langle \boldsymbol{x}, \boldsymbol{z} \rangle + Q\left(d \max_l (\boldsymbol{y})_l - \sum_{l=1}^{d} (\boldsymbol{y})_l\right), \quad (7)$$

*where $\langle ., . \rangle$ denotes the inner product operator in $\mathbb{R}^d$.*

We use Lemma 1 to prove the following theorem concerning the device heterogeneity effect on the FL convergence bound. Devices with lower computational capability partially train the global model due to the zero-valued elements in their masking vectors. We show that employing the masking vectors to address the device heterogeneity issue in FL leads to a bias term in the convergence bound. However, it does not affect the convergence rate.

**Theorem 1.** *Under Assumptions $1-3$, and for smooth and non-convex loss functions, if the total number of communication rounds $T$ is pre-defined and the learning rate $\eta(t)$ is small enough such that $\eta(t) = \eta \le \frac{1}{LN^2\tau}$, we have*

$$\frac{1}{T}\sum_{t=1}^{T} \mathbb{E}\|\nabla F(\boldsymbol{w}_g(t))\|^2 \le \frac{2}{\eta\tau T}(F(\boldsymbol{w}_g(1)) - F^*) + LN\tau\eta\sum_{n=1}^{N}\xi_n^2$$

$$+ 2\Psi\sum_{n=1}^{N}\left(d_{\boldsymbol{w}}\gamma_n - \sum_{l=1}^{d_{\boldsymbol{w}}}(\boldsymbol{k}_n)_l\right) + L^2\eta^2 G^2\frac{(\tau-1)(2\tau-1)}{6}, \quad (8)$$

*where $\Psi$ is a constant satisfying $\big|\max_l \big(\nabla f_n(\boldsymbol{w}_n^i(t)) \odot \nabla F(\boldsymbol{w}_g(t))\big)_l\big| \le \Psi$ for all $n \in [N]$, $i = 1, \ldots, \tau$, and $t = 1, \ldots, T$. $F^* = F(\boldsymbol{w}^*)$, where $\boldsymbol{w}^*$ is the global optimal point. $L$, $\xi_n^2$, and $G$ are constants defined in Assumptions $1-3$.*

The proof for Lemma 1 and Theorem 1 can be found in Appendices D and E, respectively.

**Remark 1.** *By employing the masking vectors in FL, the term $\sum_{n=1}^{N}\left(d_{\boldsymbol{w}}\gamma_n - \sum_{l=1}^{d_{\boldsymbol{w}}}(\boldsymbol{k}_n)_l\right)$ appears on the right-hand side of (8). Since this term does not scale with the number of communication rounds $T$, it is considered as a bias term, which remains as a residual in the convergence bound. Hence, those FL algorithms, which use masking vectors to reduce the computational and communication costs for the devices, may converge to a local minimum of the objective function $F(\boldsymbol{w}_g)$. In PerFedMask, we design the masking vectors by minimizing $\sum_{n=1}^{N}\left(d_{\boldsymbol{w}}\gamma_n - \sum_{l=1}^{d_{\boldsymbol{w}}}(\boldsymbol{k}_n)_l\right)$ to mitigate the performance degradation due to this bias term. In Appendix F, we present a simple example to gain insight regarding the selection of masking vectors by minimizing the bias term.*

## 5 OUR PROPOSED ALGORITHM

In this section, we propose a novel algorithm called PerFedMask, which aims to mitigate the performance degradation caused by bias described in Remark 1 through: (1) systematically designing the

---

**Algorithm 1:** Training Procedure of PerFedMask

---

1: **Input**: Local datasets $\mathcal{D}_n$; maximum number of trainable parameters $\psi_n$, $n \in [N]$; the number of local epochs $E$; the number of local batches $B$; participation ratio $c$.
2: Initialize the learning rate $\eta(1)$ and initialize randomly $\boldsymbol{\theta}_g(1) := \{\boldsymbol{w}_g(1), \boldsymbol{\phi}_g\}$.  # Model initialization
3: "**Server Operation**"
4: Select masking vector $\boldsymbol{m}_n$ for each device $n$ by solving $\mathcal{P}^{\mathrm{mask}}$. # Optimized masking using (9a)−(9d)
5: **for** each communication round $t \in \{1, \ldots, T\}$ **do**
6:     $\mathcal{N}(t) \leftarrow$ Random subset of $\max(cN, 1)$ devices.          # Selection of participating devices
7:     **for** each device $n \in \mathcal{N}(t)$ in parallel **do**
8:         $\boldsymbol{w}_n^{\tau+1}(t) = \mathrm{DeviceLocalUpdate}(\boldsymbol{w}_g(t), \boldsymbol{\phi}_g, \boldsymbol{m}_n, f_n, \mathcal{D}_n, \eta(t))$.    # Local updates using eqn. (1)
9:     **end for**
10:    $\boldsymbol{w}_g(t+1) = \sum_{n \in \mathcal{N}(t)} \boldsymbol{k}_n \odot \boldsymbol{w}_n^{\tau+1}(t)$.              # Aggregation at the server using eqn. (3)
11:    Update $\eta(t)$.
12: **end for**
13: "**Client Operation**"
14: **for** each device $n \in [N]$ in parallel **do**
15:    Fine-tune the learning model $\boldsymbol{\theta}_n := \{\boldsymbol{w}_g(T+1), \boldsymbol{\phi}_n\}$ using training data samples $\mathcal{D}_n$.  # Fine-tuning
16: **end for**

---

masking vectors via an optimization framework, and (2) fine-tuning the local head models. First, each device determines the maximum number of parameters which can be trained during local update iterations based on its computational capability[2]. These values are sent to the server. The server then determines the masking vector for each device before training.

Let $\psi_n$ denote the maximum number of parameters that can be trained by device $n \in [N]$, where $\psi_n \leq d_{\boldsymbol{w}}$. Given $\psi_n$, the server determines the masking vector $\boldsymbol{m}_n$ for each device $n$ before training by minimizing the bias term. Here, we present the formulation of layer-wise masking[3], which decides whether or not to freeze all the parameters in each layer of the global model. We adopt layer-wise masking since most of the current machine learning frameworks such as PyTorch (PyTorch, 2022) and TensorFlow (Abadi et al., 2016) run at the granularity of a full tensor, with no APIs which can provide parameter freezing at a finer granularity. Also, by considering layer-wise masking, the number of optimization variables can be reduced. Thus, the complexity of obtaining the masking vectors can also be reduced accordingly. Let $\Lambda$ and $|\Lambda|$ denote the set and number of layers in the global model, respectively. Let $\pi_j$ and $|\pi_j|$ denote the set and number of parameters in layer $j \in \Lambda$, respectively. We define $\tilde{\boldsymbol{m}}_n \in \{0, 1\}^{|\Lambda|}$ as the layer-wise masking vector. If $(\tilde{\boldsymbol{m}}_n)_j = 1$, then all the elements $(\boldsymbol{m}_n)_l$, $l \in \pi_j$ are equal to one. In PerFedMask, the following optimization problem is solved by the server to obtain the masking vectors for the devices:

$$\mathcal{P}^{\mathrm{mask}}: \minimize_{\tilde{\boldsymbol{m}}_n, \epsilon_n, n \in [N]} \quad \sum_{n=1}^{N} \left( d_{\boldsymbol{w}} \max_{j \in \Lambda} (\tilde{\boldsymbol{k}}_n)_j - \sum_{j' \in \Lambda} |\pi_{j'}| (\tilde{\boldsymbol{k}}_n)_{j'} + \epsilon_n \right)$$

$$\text{subject to} \quad (\tilde{\boldsymbol{k}}_n)_j = \frac{(\tilde{\boldsymbol{m}}_n)_j}{\sum_{n'=1}^{N} (\tilde{\boldsymbol{m}}_{n'})_j}, \; j \in \Lambda, \; n \in [N], \tag{9a}$$

$$\sum_{j \in \Lambda} |\pi_j| (\tilde{\boldsymbol{m}}_n)_j = \psi_n - \epsilon_n, \; n \in [N], \tag{9b}$$

$$(\tilde{\boldsymbol{m}}_n)_j \in \{0, 1\}, \; j \in \Lambda, \; n \in [N], \tag{9c}$$

$$\epsilon_n \geq 0, \; n \in [N], \tag{9d}$$

where $\epsilon_n$ is a slack variable, which prevents to train more than $\psi_n$ parameters for each device $n$ in the layer-wise masking. Problem $\mathcal{P}^{\mathrm{mask}}$ is a mixed-integer nonlinear program, which is NP-hard and difficult to solve. In Appendix H, we show how to obtain a close-to-optimal solution by using successive convex approximation (Shen et al., 2016) and MOSEK solver (MOSEK ApS, 2022). Algorithm 1 summarizes the training procedure of PerFedMask.

---

[2]Using the curves similar to those in Fig. 3 in Appendix G, each device can determine its maximum number of trainable parameters.

[3]Random layer-wise masking has been considered in Sidahmed et al. (2021); Pfeiffer et al. (2022).

## 6 EXPERIMENTAL RESULTS

### 6.1 EXPERIMENT SETUP

**Datasets and Model Architectures.** We conduct our experiments on CIFAR-10 and CIFAR-100 image classification tasks[4]. Our experiments are performed with ResNet (PreResNet18) (He et al., 2016) for CIFAR-10, and with MobileNet (Howard et al., 2017) for CIFAR-100. In both cases, we set the number of devices to 100. The data samples are uniformly divided among the devices. Each device has 450 training data samples, 50 validation data samples, and 100 test data samples. The batch size is set to 50. To enable non-IID data partitioning among the devices, we distribute 3 and 10 classes per device for CIFAR-10 and CIFAR-100 datasets, respectively. The same classes are considered in the training, validation, and test datasets.

**Implementation Details.** We denote the fraction of devices with the maximum computational capability by $\nu$. That is, $\nu$ represents the ratio of devices which can completely update the entire global model during the local update iterations. Those devices are able to train $\times 1$ parameters of the global model. Since the remaining devices have lower computational capability, they should mask some parts of the global model during the local update iterations based on their capabilities. Unless stated otherwise, we set $\nu = 0.5$. For all the experiments, the learning rate starts with $0.1$ and is decayed by a factor of $0.1$ in communication round $t \in \{\frac{1}{2}T, \frac{3}{4}T\}$. Similar to FedBABU (Oh et al., 2022), we fix the product of the local epochs $E$ and the maximum number of communication rounds $T$ to 320. After $T$ communication rounds, we choose the model with the maximum validation accuracy and perform 5 local update iterations for fine-tuning the learning model using each device's local training data samples. We perform the experiments using PyTorch library (PyTorch, 2022) in Python 3.7. We apply layer-wise masking in our experiments.

**Baselines.** We compare the performance of our proposed algorithm, PerFedMask, with the following FL algorithms: FedBABU (Oh et al., 2022) and FedProx (Li et al., 2020a), which have been proposed to tackle data heterogeneity; FedNova (Wang et al., 2020), which has been proposed to address objective inconsistency problem; HeteroFL (Diao et al., 2021) and Split-Mix FL (Hong et al., 2022), which have been proposed to tackle device heterogeneity in the non-IID data settings.

**Performance Metrics.** We consider the test accuracy as one of the performance metrics. Fine-tuning steps are performed for PerFedMask and FedBABU algorithms to personalize the learning model and obtain the local head model for each device. For fair comparison, the obtained learning model has been also fine-tuned for each device in the other FL algorithms. Thus, we report the test accuracy before and after fine-tuning. We also assess the average number of floating-point operations (FLOPs) in each communication round for both the forward and backward propagation paths to show the required computation for the algorithms. We use PyPAPI (PyPAPI, 2017) to obtain the number of FLOPs. Since the average number of trainable parameters is equal to the average number of parameters transmitted from the devices to the server in each communication round, we report this number to show the communication cost of each FL algorithm.

### 6.2 BENCHMARK EXPERIMENTS

We consider a heterogeneous device setting, where half of the devices (i.e., devices with maximum computational capability) perform four local epochs (i.e., $E = 4$). Due to the limited computational capability, the remaining devices perform two local epochs for updating all the parameters. Using PerFedMask, instead of considering different local epochs for the devices, we address the device heterogeneity issue by freezing some parts of the global model for those devices with lower computational capability. For fair comparison, the number of frozen parameters are selected in a way such that the considered algorithms have the same number of FLOPs for each device.

**Performance Comparison with the Baselines.** Table 1 shows the obtained test accuracy after fine-tuning and the number of trainable parameters for CIFAR-10 and CIFAR-100 datasets [5]. Our observations are as follows: (1) our proposed algorithm, PerFedMask, has comparable performance

---

[4]We also use AlexNet on DomainNet dataset (Li et al., 2021) and provide the results in Appendix I to show the performance under feature non-IID configuration.

[5]We show some of the training curves in Appendix J. Also, in Appendix K, we present additional statistics for some of the results in Table 1.

Table 1: Test accuracy after fine-tuning and number of trainable parameters of PerFedMask and the baseline algorithms for CIFAR-10 and CIFAR-100 datasets

| | | **Test accuracy after fine-tuning** | | | | | | |
|---|---|---|---|---|---|---|---|---|
| Dataset | $c$ | PerFedMask (Ours) | FedBABU | FedProx | FedNova | HeteroFL | Split-Mix FL | FedAvg |
| CIFAR-10 | 1 | **88.43** | 88.20 | 84.96 | 84.26 | 87.33 | 85.56 | 84.99 |
| | 0.1 | 83.60 | **84.27** | 74.55 | 71.88 | 73.34 | 77.76 | 71.19 |
| CIFAR-100 | 1 | **72.40** | 69.01 | 64.63 | 65.24 | 68.65 | 65.95 | 65.27 |
| | 0.1 | **67.47** | 66.32 | 59.36 | 60.42 | 65.87 | 62.35 | 59.12 |
| | | **Number of trainable parameters** | | | | | | |
| Dataset | | PerFedMask (Ours) | FedBABU | FedProx | FedNova | HeteroFL | Split-Mix FL | FedAvg |
| CIFAR-10 | | 6.138M | 11.167M | 11.172M | 11.172M | 5.674M | 0.793M | 11.172M |
| CIFAR-100 | | 1.803M | 3.207M | 3.309M | 3.309M | 1.774M | 0.223M | 3.309M |

to FedBABU and outperforms the other baselines in terms of test accuracy after fine-tuning. (2) By increasing the number of devices participating in FL (i.e., by increasing $c$), a higher test accuracy can be achieved. (3) PerFedMask, HeteroFL, and Split-Mix FL can provide lower number of trainable parameters. Split-Mix FL has the lowest number of trainable parameters because it trains several low-width base models instead of the original learning model. (4) Different number of local update iterations for the devices may lead to the objective inconsistency problem. PerFedMask, HeteroFL, and Split-Mix FL algorithms can address this problem by decreasing the number of FLOPs for the less capable devices. Thus, the same number of local update iterations can be considered for all the devices. Also, this problem is tackled in FedNova through modifying the optimizer. However, other algorithms suffer from the objective inconsistency problem, which may degrade their performance.

**Combining PerFedMask with HeteroFL and Split-Mix FL.** Table 2 shows the performance results for PerFedMask and its combination with HeteroFL and Split-Mix FL algorithms. Table 2 also shows the performance results for FedBABU, HeteroFL, and Split-Mix FL algorithms. The number of FLOPs for forward and backward indicates the number of required computations in the forward and backward propagation paths, respectively. In general, backpropagation dominates the computational cost during training of a learning model (Xu et al., 2022). In PerFedMask, by using masking vectors, there is no need to compute the partial derivative of the objective function with respect to the frozen parameters. Although PerFedMask has reduced the number of trainable parameters and the backward FLOPs, it can still achieve higher test accuracy compared to other algorithms including FedBABU. Since PerFedMask does not change the architecture of the learning model, it can easily be combined with other FL algorithms. Table 2 shows that combining PerFedMask with Split-Mix FL and HeteroFL algorithms can further reduce the number of FLOPs in the backward path and the number of trainable parameters. This combination provides a higher test accuracy after fine-tuning than Split-Mix FL and HeteroFL algorithms.

Table 2: Performance comparison on CIFAR-10 dataset when $c = 1$. Results for CIFAR-100 dataset can be found in Appendix L.

| Algorithm | Test accuracy | | # of trainable parameters | # of FLOPs | |
|---|---|---|---|---|---|
| | Before fine-tuning | After fine-tuning | | Forward | Backward |
| PerFedMask + Split-Mix FL | 51.88 | 87.74 | 0.691M | 0.178G | 0.514G |
| PerFedMask + HeteroFL | 69.44 | 87.79 | 5.473M | 1.111G | 1.721G |
| PerFedMask | 70.14 | 88.43 | 6.138M | 2.182G | 2.697G |
| Split-Mix FL | 57.96 | 85.56 | 0.793M | 0.178G | 0.541G |
| HeteroFL | 62.58 | 87.33 | 5.674M | 1.111G | 1.749G |
| FedBABU | 69.27 | 88.20 | 11.167M | 2.182G | 3.466G |

## 6.3 ABLATION STUDIES

**Effect of Fine-Tuning Steps.** We investigate the impact of the number of fine-tuning steps on the final test accuracy of PerFedMask and FedBABU [6]. Fine-tuning steps equal to zero means the test accuracy is obtained before fine-tuning. Also, since the number of batches for each device's training dataset is equal to 9, increasing the fine-tuning steps by one leads to 9 more local update iterations. Results from Table 3 show that similar to FedBABU, PerFedMask can achieve better accuracy with

---

[6]We also investigate the effect of freezing the local head models in Appendix M.

a small number of fine-tuning steps. This characteristic is important when fine-tuning is restricted or costly for the devices.

Table 3: Performance according to fine-tuning steps when $c = 1$

| Dataset | Algorithm | Fine-tuning steps | | | | | |
|---|---|---|---|---|---|---|---|
| | | 0 | 2 | 4 | 6 | 8 | 10 |
| CIFAR-10 | PerFedMask | 70.14 | 88.78 | 88.69 | 88.35 | 88.25 | 88.30 |
| | FedBABU | 69.27 | 88.63 | 88.38 | 88.43 | 88.12 | 87.92 |
| CIFAR-100 | PerFedMask | 32.04 | 72.62 | 72.84 | 72.48 | 72.11 | 72.40 |
| | FedBABU | 29.70 | 69.03 | 69.15 | 68.86 | 68.83 | 68.81 |

**Effect of Increasing the Number of Devices with Maximum Computational Capability.** We investigate the impact of increasing $\nu$ on the performance of PerFedMask in Table 4. More devices in the network are able to train the entire global model as $\nu$ increases. This leads to an increase in the number of trainable parameters and number of backward FLOPs in PerFedMask. We can observe that by increasing $\nu$, the test accuracy before fine-tuning is improved. Note that PerFedMask can provide a comparable test accuracy after fine-tuning even for $\nu = 0.2$, when compared with the case in which all devices have the maximum computational capability (i.e., $\nu = 1$).

Table 4: Results of increasing $\nu$ for CIFAR-100 dataset when $c = 1$.

| Algorithm | $\nu$ | Test accuracy | | # of trainable parameters | # of backward FLOPs |
|---|---|---|---|---|---|
| | | Before fine-tuning | After fine-tuning | | |
| PerFedMask | 0.2 | 29.29 | 72.07 | 0.941M | 0.617G |
| | 0.4 | 32.31 | 74.33 | 1.518M | 0.675G |
| | 0.6 | 32.79 | 72.82 | 2.095M | 0.741G |
| | 0.8 | 33.59 | 72.64 | 2.647M | 0.803G |
| | 1.0 | 34.73 | 73.76 | 3.207M | 0.863G |

**Effect of Masking Vectors Design.** In Table 5, we consider three masking approaches: in sequential masking, the layers are masked sequentially; in random masking, the layers are masked randomly; and in optimized masking, the layers are masked by solving problem $\mathcal{P}^{\text{mask}}$. Optimized masking minimizes the bias described in Remark 1. As shown in Table 5, optimized masking can provide lower training loss and higher training and test accuracies. These results are compatible with our theoretical analysis. As shown in Table 5, PerFedMask enhances FL performance before fine-tuning by employing optimal masking vectors. The final test accuracy can then be improved by fine-tuning.

Table 5: Results of different approaches for masking vectors design for CIFAR-100 dataset when $c = 1$ and $\nu = 0.2$.

| Design approach | Bias | Training loss | Training accuracy | Test accuracy before fine-tuning |
|---|---|---|---|---|
| Sequential masking | 3.364M | 2.324 | 50.11 | 27.02 |
| Random masking | 0.609M | 1.706 | 64.40 | 28.88 |
| Optimized masking | 0.204M | 1.648 | 66.63 | 29.84 |

## 7 CONCLUSION

In this work, we proposed a flexible and easy to implement personalized FL algorithm called PerFed-Mask. We provided theoretical and empirical grounds to justify the utility of PerFedMask in heterogeneous data and device settings. In particular, PerFedMask employs (1) optimized masking vectors obtained by minimizing the bias term in the convergence bound, and (2) fine-tuning. The masking vectors are exploited to freeze some parts of the global model for each device based on its computational capability. Fine-tuning is performed by each device after training to improve the final test accuracy. When compared with some existing state-of-the-art FL algorithms, PerFedMask can achieve higher test accuracy. It can also decrease the average number of trainable parameters and the average number of FLOPs in each communication round without changing the learning model architecture. A future direction is to consider freezing priority for different layers in the neural network architecture. For example, in Frankle et al. (2021), it has been shown that batch normalization layers in convolutional networks are important to be considered as the trainable parameters. Also, the approach of masking vectors design in this work can be emulated in pruning methods.

ACKNOWLEDGMENTS

This work was supported in part by Rogers Communications Canada Inc., Natural Sciences and Engineering Research Council of Canada (NSERC), and Public Safety Canada (NS-5001-22170).

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

# A  RELATED WORK

**FL Algorithms with non-IID Data.** In FedAvg (McMahan et al., 2017), the edge devices perform multiple local update iterations on their local models before sending them to the server for aggregation. In this way, the communication cost can be reduced. When there is data heterogeneity, conventional FL algorithms such as FedAvg may have slow and unstable convergence (Karimireddy et al., 2020). To handle the case when the data is non-IID, some works (Li et al., 2020a; Karimireddy et al., 2020) have introduced new optimization frameworks to obtain a more stable model for the devices. Although the obtained model may still perform well on average across the devices, some of the devices may not be satisfied with the obtained final accuracy (Collins et al., 2021). To address this issue, more personalized models are required for the devices. The personalized models can be obtained by using different methods such as meta-learning (Fallah et al., 2020), multi-task learning (Marfoq et al., 2021), clustering (Mansour et al., 2020), and decomposition. In the decomposition method, the learning model is decomposed into a global model and a device-specific head model (Arivazhagan et al., 2019; Collins et al., 2021). In each communication round, the global model is updated by the devices and is aggregated by the server, whereas the head model is updated by each device but is not transmitted to the server. Oh et al. (2022) proposed FedBABU using the decomposition method. To improve the personalization ability, in FedBABU, the local head model parameters are frozen during training. After convergence to a global model, each device updates the local head model parameters during the fine-tuning steps by using its local data samples.

**FL Algorithms with Heterogeneous Devices.** Given the disparities in devices' hardware, it is crucial to address device heterogeneity in FL. In general, this problem can be tackled by reducing the computational complexity of model training based on the devices' hardware capabilities. Lin et al. (2020); Afonin & Karimireddy (2022) proposed FL algorithms using knowledge distillation, which aim to generate compact models by transferring knowledge of a large model to smaller ones. Pruning method, where a compact model for each device can be obtained by removing the parameters with little impact on the performance of the original learning model, is another approach to accommodate device heterogeneity in FL. Due to the dynamic sparse training, pruning method may suffer from instability in convergence for finding the sparse masking vectors, which are used for training the sparse sub-networks at the devices. This issue can be resolved by consistent mask adjustment procedure at the devices, at the expense of additional communication overhead (Jiang et al., 2022; Babakniya et al., 2022; Bibikar et al., 2022). The ordered dropout technique has been proposed in Horvath et al. (2021) to dynamically adapt the model size used by each device based on its capabilities. Another pruning method that does not require additional communication cost is to use a fixed masking vector for each device before training based on its computational capability. This would result in training of the heterogeneous local models for the devices. In this regard, HeteroFL (Diao et al., 2021) aims to facilitate efficient training and stable aggregation of devices' local models. However, the available local data samples at the devices may not be used efficiently in HeteroFL. In particular, HeteroFL considers different model architectures as the local models for the devices. Thereby, the available data samples at each device can be used only for training a specific model architecture. In Split-Mix FL (Hong et al., 2022), based on its computational capability, each device randomly selects some of the base models in each communication round to train them. After training, by using ensemble learning approach, each device mixes the selected base models to construct its desired model size. The potential drawback of this approach is that none of the devices train the original learning model.

Another line of research is related to the objective inconsistency problem in FL. Due to the different size of the devices' local dataset and their computational capability, some of the devices may finish their local update iterations faster. To prevent the devices from being idle, Wang et al. (2020) proposed to let those faster devices continue their local update iterations until the slowest device finishes its local update. FedNova is proposed by Wang et al. (2020) to resolve the objective inconsistency problem due to different number of local update iterations. Another approach is to consider the same number of local update iterations for all the devices. However, to enable all the devices to finish their local update iterations in a timely manner, the required number of computations should be adjusted for those devices with lower computational capabilities. Our proposed algorithm as well as other FL algorithms which aim to adjust the required computations for each device based on its computational capability can be employed to address the objective inconsistency problem.

# B  LOCAL UPDATES AT DEVICES

---

**Algorithm 2:** Local Update Function

---
1: **function** DeviceLocalUpdate($\boldsymbol{w}_g, \boldsymbol{\phi}_g, \boldsymbol{m}, f, \mathcal{D}, \eta$)     # Local update iterations for each device
2:    $\mathcal{B} \leftarrow$ Split local data samples $\mathcal{D}$ into $B$ local batches.
3:    $i := 1$, and $\boldsymbol{w}^i \leftarrow \boldsymbol{w}_g$.
4:    **for** each local epoch $e \in \{1, \ldots, E\}$ **do**     # The total number of local update iterations $\tau = E \times B$
5:       **for** each batch $b \in \mathcal{B}$ **do**
6:          $\boldsymbol{w}^{i+1} \leftarrow \boldsymbol{w}^i - \eta \boldsymbol{m} \odot \nabla f(\boldsymbol{w}^i, b)$.
7:          $i := i + 1$.
8:       **end for**
9:    **end for**
10:    **Return** $\boldsymbol{w}^i$
11: **end function**

---

# C  CONVERGENCE ANALYSIS FOR STRONGLY CONVEX LOSS FUNCTIONS

To show the convergence of PerFedMask for smooth and strongly convex loss functions, in addition to Assumptions $1-3$, we make the following assumption:

**Assumption 4.** $f_n(\boldsymbol{w})$, $n \in [N]$, is $\mu$-strongly convex and satisfies:

$$f_n(\boldsymbol{v}) \geq f_n(\boldsymbol{w}) + (\boldsymbol{v} - \boldsymbol{w})^T \nabla f_n(\boldsymbol{w}) + \frac{\mu}{2} \|\boldsymbol{v} - \boldsymbol{w}\|^2, \quad \forall \boldsymbol{v}, \boldsymbol{w}, \ n \in [N]. \tag{10}$$

To quantify the degree of data heterogeneity at each device $n \in [N]$, the term $\Gamma_n = f_n(\boldsymbol{w}^*) - f_n^*$ is defined (Li et al., 2020b). Let $\delta(t) = \mathbb{E}\|\boldsymbol{w}_g(t) - \boldsymbol{w}^*\|^2$. We first prove the following useful lemma.

**Lemma 2.** Under Assumptions $1-4$, if the learning rate is small enough, i.e., $\eta(t) \leq \frac{1}{L(N\tau+1)}$, for all $t = 1, \ldots, T$, we have

$$\delta(t + 1) \leq (1 - q_0 \, \eta(t)) \, \delta(t) + q_1 \, \eta(t) + q_2 \, \eta^2(t), \tag{11}$$

where

$$q_0 \triangleq \frac{1}{2}\mu\tau \sum_{n=1}^{N} \gamma_n, \tag{12}$$

$$q_1 \triangleq 2\tau\Upsilon \sum_{n=1}^{N} \left(\gamma_n - \frac{1}{N}\right) + 2\tau\Omega \sum_{n=1}^{N} \left(d_{\boldsymbol{w}}\gamma_n - \sum_{l=1}^{d_{\boldsymbol{w}}} (\boldsymbol{k}_n)_l\right), \tag{13}$$

$$q_2 \triangleq G^2 \frac{\tau(\tau - 1)(2\tau - 1)}{6} (2 + \mu) \sum_{n=1}^{N} \gamma_n + 2L\tau \sum_{n=1}^{N} \gamma_n (2 + N\tau\gamma_n) \Gamma_n + N\tau^2 \sum_{n=1}^{N} \xi_n^2, \tag{14}$$

and $\Omega$ and $\Upsilon$ should satisfy $\left|\min_l \left((\boldsymbol{w}^* - \boldsymbol{w}_n^i(t)) \odot \nabla f_n(\boldsymbol{w}_n^i(t))\right)_l\right| \leq \Omega$, for all $n \in [N]$, $i = 1, \ldots, \tau$, $t = 1, \ldots, T$ and $\left|\min_{n \in [N]} (f_n(\boldsymbol{w}_g(t)) - f_n(\boldsymbol{w}^*))\right| \leq \Upsilon$, $t = 1, \ldots, T$, respectively.

*Proof.* See Appendix N.  □

Using Lemma 2, we can state the following theorem for the convergence rate of smooth and strongly convex loss functions:

**Theorem 2.** Given Assumptions $1-4$, if we choose $\kappa = \frac{2L}{q_0}(N\tau + 1)$ and the learning rate $\eta(t) = \frac{2}{q_0(t+\kappa)}$, under the full device participation scenario (i.e., $c = 1$), after $T$ communication rounds, we have

$$\mathbb{E}F(\boldsymbol{w}_g(T)) - F^* \leq \frac{Lq_1}{q_0} + \frac{L}{2(T + \kappa)} \left(\frac{4q_2}{q_0^2} + (\kappa + 1)\delta(1)\right). \tag{15}$$

*Proof.* See Appendix O. □

**Remark 2.** *The first term on the right-hand side of (15) appears in the convergence bound due to the device heterogeneity. In particular, Theorem 2 shows that for $q_1 \neq 0$, the FL algorithm converges to a local optimal solution at the rate of $O(1/T)$. This convergence rate is similar to the results presented in Li et al. (2020b); Amiri et al. (2022), where device heterogeneity has not been considered. Hence, using the masking vectors in FL do not degrade the convergence rate of FL.*

**Remark 3.** *The result in Theorem 2 shows that when $q_1 \to 0$, the FL algorithm converges to the global optimal solution for the smooth and strongly convex functions. Based on (13), it is straightforward to verify that without device heterogeneity in the network (i.e., when all the devices have the maximum computational capability and can update all the parameters of the global model), all the elements of vector $\boldsymbol{k}_n$, $n \in [N]$ are equal to $\frac{1}{N}$. Hence, in this case, we have $\gamma_n = \frac{1}{N}$, and $q_1 = 0$. In general, based on (13), one way to reduce the bias incurred by the device heterogeneity is to design the masking vectors by minimizing $q_1$. For example, one can search for the masking vectors, which minimize $\sum_{n=1}^{N} \left( d_{\boldsymbol{w}} \gamma_n - \sum_{l=1}^{d_{\boldsymbol{w}}} (\boldsymbol{k}_n)_l \right)$.*

## D   PROOF OF LEMMA 1

Given vectors $\boldsymbol{x}$, $\boldsymbol{y}$, and $\boldsymbol{z}$, we form diagonal matrices $\boldsymbol{X}$, $\boldsymbol{Y}$, and $\boldsymbol{Z}$, respectively. Note that we can write $\langle \boldsymbol{x}, \boldsymbol{y} \odot \boldsymbol{z} \rangle$ as the form of the trace of matrices $\boldsymbol{X}$, $\boldsymbol{Y}$, and $\boldsymbol{Z}$ product, i.e., $\langle \boldsymbol{x}, \boldsymbol{y} \odot \boldsymbol{z} \rangle = \mathrm{Tr}(\boldsymbol{XYZ})$. By using Theorem 3 in Fang et al. (1994), we have the following inequality:

$$\mathrm{Tr}(\boldsymbol{XYZ}) \leq \lambda_1(\boldsymbol{Y}) \mathrm{Tr}(\boldsymbol{XZ}) - \lambda_d(\boldsymbol{XZ}) \left( d\lambda_1(\boldsymbol{Y}) - \mathrm{Tr}(\boldsymbol{Y}) \right), \tag{16}$$

where $\lambda_1(\boldsymbol{Y})$ and $\lambda_d(\boldsymbol{XZ})$ are the largest eigenvalue of matrix $\boldsymbol{Y}$ and the smallest eigenvalue of matrix $\boldsymbol{XZ}$, respectively. Since the considered matrices are diagonal, we have $\lambda_1(\boldsymbol{Y}) = \max_l (\boldsymbol{y})_l$ and $\lambda_d(\boldsymbol{XZ}) = \min_l (\boldsymbol{x} \odot \boldsymbol{z})_l$. Hence, we have

$$\langle \boldsymbol{x}, \boldsymbol{y} \odot \boldsymbol{z} \rangle \leq \max_l (\boldsymbol{y})_l \langle \boldsymbol{x}, \boldsymbol{z} \rangle - \min_l (\boldsymbol{x} \odot \boldsymbol{z})_l \left( d \max_l (\boldsymbol{y})_l - \sum_{l=1}^{d} (\boldsymbol{y})_l \right). \tag{17}$$

Since $d \max_l (\boldsymbol{y})_l - \sum_{l=1}^{d} (\boldsymbol{y})_l \geq 0$, by considering $\min_l (\boldsymbol{x} \odot \boldsymbol{z})_l \geq -Q$, Lemma 1 is proved using inequality (17).

## E   PROOF OF THEOREM 1

Considering the smoothness of $f_n(\boldsymbol{w})$, $n \in [N]$, in each communication round $t \geq 1$, we have

$$\mathbb{E}F(\boldsymbol{w}_g(t+1)) \leq$$

$$\mathbb{E}F(\boldsymbol{w}_g(t)) + \mathbb{E} \langle \boldsymbol{w}_g(t+1) - \boldsymbol{w}_g(t), \nabla F(\boldsymbol{w}_g(t)) \rangle + \frac{L}{2} \mathbb{E} \|\boldsymbol{w}_g(t+1) - \boldsymbol{w}_g(t)\|^2. \tag{18}$$

We first find an upper bound for $\|\boldsymbol{w}_g(t+1) - \boldsymbol{w}_g(t)\|^2$ as follows:

$$\mathbb{E} \|\boldsymbol{w}_g(t+1) - \boldsymbol{w}_g(t)\|^2 \overset{(a)}{=} \eta^2(t) \mathbb{E} \left\| \sum_{n=1}^{N} \boldsymbol{k}_n \odot \sum_{i=1}^{\tau} \nabla f_n(\boldsymbol{w}_n^i(t), b_n^i(t)) \right\|^2$$

$$\overset{(b)}{=} \eta^2(t) \mathbb{E} \underbrace{\left\| \sum_{n=1}^{N} \sum_{i=1}^{\tau} \boldsymbol{k}_n \odot \nabla f_n(\boldsymbol{w}_n^i(t), b_n^i(t)) - \boldsymbol{k}_n \odot \nabla f_n(\boldsymbol{w}_n^i(t)) \right\|^2}_{A_1}$$

$$+ \eta^2(t) \underbrace{\left\| \sum_{n=1}^{N} \sum_{i=1}^{\tau} \boldsymbol{k}_n \odot \nabla f_n(\boldsymbol{w}_n^i(t)) \right\|^2}_{A_2}, \tag{19}$$

where equality (a) results from (2) and (3). Equality (b) is obtained via basic equality $\mathbb{E} \|\boldsymbol{z}\|^2 = \mathbb{E} \|\boldsymbol{z} - \mathbb{E}\boldsymbol{z}\|^2 + \|\mathbb{E}\boldsymbol{z}\|^2$ for any random vector $\boldsymbol{z}$.

By using Assumption 2, we can obtain an upper bound of $A_1$ as follows:

$$A_1 = \mathbb{E}\left\|\sum_{n=1}^{N}\sum_{i=1}^{\tau} \boldsymbol{k}_n \odot \nabla f_n(\boldsymbol{w}_n^i(t), b_n^i(t)) - \boldsymbol{k}_n \odot \nabla f_n(\boldsymbol{w}_n^i(t))\right\|^2$$

$$\leq N\tau \sum_{n=1}^{N}\sum_{i=1}^{\tau} \mathbb{E}\left\|\boldsymbol{k}_n \odot \nabla f_n(\boldsymbol{w}_n^i(t), b_n^i(t)) - \boldsymbol{k}_n \odot \nabla f_n(\boldsymbol{w}_n^i(t))\right\|^2$$

$$\leq N\tau^2 \sum_{n=1}^{N} \xi_n^2. \tag{20}$$

By considering the convexity of $\|\cdot\|^2$ and by using $\gamma_n = \max_l (\boldsymbol{k}_n)_l$, we can obtain an upper bound of $A_2$ as follows:

$$A_2 = \left\|\sum_{n=1}^{N}\sum_{i=1}^{\tau} \boldsymbol{k}_n \odot \nabla f_n(\boldsymbol{w}_n^i(t))\right\|^2$$

$$\leq N\tau \sum_{n=1}^{N}\sum_{i=1}^{\tau} \left\|\boldsymbol{k}_n \odot \nabla f_n(\boldsymbol{w}_n^i(t))\right\|^2$$

$$\leq N\tau \sum_{n=1}^{N}\sum_{i=1}^{\tau} \gamma_n^2 \left\|\nabla f_n(\boldsymbol{w}_n^i(t))\right\|^2. \tag{21}$$

By combining (19), (20), and (21), we have the following inequality:

$$\mathbb{E}\left\|\boldsymbol{w}_g(t+1) - \boldsymbol{w}_g(t)\right\|^2 \leq N\tau^2\eta^2(t)\sum_{n=1}^{N}\xi_n^2 + N\tau\eta^2(t)\sum_{n=1}^{N}\sum_{i=1}^{\tau}\gamma_n^2\left\|\nabla f_n(\boldsymbol{w}_n^i(t))\right\|^2. \tag{22}$$

Now, we aim to obtain an upper bound of $\mathbb{E}\langle \boldsymbol{w}_g(t+1) - \boldsymbol{w}_g(t), \nabla F(\boldsymbol{w}_g(t))\rangle$. We have

$$\mathbb{E}\langle \boldsymbol{w}_g(t+1) - \boldsymbol{w}_g(t), \nabla F(\boldsymbol{w}_g(t))\rangle$$

$$\stackrel{(a)}{=} \mathbb{E}\left\langle -\eta(t)\sum_{n=1}^{N}\sum_{i=1}^{\tau}\boldsymbol{k}_n \odot \nabla f_n(\boldsymbol{w}_n^i(t), b_n^i(t)), \nabla F(\boldsymbol{w}_g(t))\right\rangle$$

$$\stackrel{(b)}{=} \eta(t)\mathbb{E}\sum_{n=1}^{N}\sum_{i=1}^{\tau}\left\langle \boldsymbol{k}_n \odot \nabla f_n(\boldsymbol{w}_n^i(t)), -\nabla F(\boldsymbol{w}_g(t))\right\rangle$$

$$\stackrel{(c)}{\leq} \eta(t)\mathbb{E}\sum_{n=1}^{N}\sum_{i=1}^{\tau}(-\gamma_n)\left\langle \nabla f_n(\boldsymbol{w}_n^i(t)), \nabla F(\boldsymbol{w}_g(t))\right\rangle + \eta(t)\tau\Psi\sum_{n=1}^{N}\left(d_{\boldsymbol{w}}\gamma_n - \sum_{l=1}^{d_{\boldsymbol{w}}}(\boldsymbol{k}_n)_l\right)$$

$$\stackrel{(d)}{\leq} -\eta(t)\sum_{i=1}^{\tau}\mathbb{E}\left\langle \frac{1}{N}\sum_{n=1}^{N}\nabla f_n(\boldsymbol{w}_n^i(t)), \nabla F(\boldsymbol{w}_g(t))\right\rangle + \eta(t)\tau\Psi\sum_{n=1}^{N}\left(d_{\boldsymbol{w}}\gamma_n - \sum_{l=1}^{d_{\boldsymbol{w}}}(\boldsymbol{k}_n)_l\right), \tag{23}$$

where equality (a) results from (2) and (3). Equality (b) follows from $\mathbb{E}\nabla f_n(\boldsymbol{w}_n^i(t), b_n^i(t)) = \nabla f_n(\boldsymbol{w}_n^i(t))$. Inequality (c) holds by using Lemma 1. Inequality (d) follows from $\gamma_n \geq \frac{1}{N}$.

To find an upper bound for $-\mathbb{E}\left\langle \frac{1}{N}\sum_{n=1}^{N}\nabla f_n(\boldsymbol{w}_n^i(t)), \nabla F(\boldsymbol{w}_g(t))\right\rangle$, we first represent it as follows:

$$-\mathbb{E}\left\langle \frac{1}{N}\sum_{n=1}^{N}\nabla f_n(\boldsymbol{w}_n^i(t)), \nabla F(\boldsymbol{w}_g(t))\right\rangle$$

$$= \frac{1}{2}\mathbb{E}\left\|\frac{1}{N}\sum_{n=1}^{N}\left(\nabla f_n(\boldsymbol{w}_n^i(t)) - \nabla f_n(\boldsymbol{w}_g(t))\right)\right\|^2$$

$$- \frac{1}{2}\mathbb{E}\left\|\frac{1}{N}\sum_{n=1}^{N}\nabla f_n(\boldsymbol{w}_n^i(t))\right\|^2 - \frac{1}{2}\mathbb{E}\left\|\nabla F(\boldsymbol{w}_g(t))\right\|^2. \tag{24}$$

Then, $\mathbb{E}\left\|\frac{1}{N}\sum_{n=1}^{N}\left(\nabla f_n(\boldsymbol{w}_n^i(t)) - \nabla f_n(\boldsymbol{w}_g(t))\right)\right\|^2$ is bounded as follows:

$$\mathbb{E}\left\|\frac{1}{N}\sum_{n=1}^{N}\left(\nabla f_n(\boldsymbol{w}_n^i(t)) - \nabla f_n(\boldsymbol{w}_g(t))\right)\right\|^2 \overset{(a)}{\leq} \frac{1}{N}\sum_{n=1}^{N}\mathbb{E}\left\|\nabla f_n(\boldsymbol{w}_g(t)) - \nabla f_n(\boldsymbol{w}_n^i(t))\right\|^2$$

$$\overset{(b)}{\leq} \frac{L^2}{N}\sum_{n=1}^{N}\mathbb{E}\left\|\boldsymbol{w}_g(t) - \boldsymbol{w}_n^i(t)\right\|^2, \tag{25}$$

where inequality (a) results from the convexity of $\|\cdot\|^2$. Inequality (b) results from Assumption 1. Now, we aim to bound $\mathbb{E}\left\|\boldsymbol{w}_g(t) - \boldsymbol{w}_n^i(t)\right\|^2$ for $i = 2,\dots,\tau$. By using (1), we have

$$\mathbb{E}\left\|\boldsymbol{w}_g(t) - \boldsymbol{w}_n^i(t)\right\|^2 = \mathbb{E}\left\|\eta(t)\boldsymbol{m}_n \odot \sum_{j=1}^{i-1}\nabla f_n(\boldsymbol{w}_n^j(t), b_n^j(t))\right\|^2$$

$$\leq \eta^2(t)(i-1)\sum_{j=1}^{i-1}\mathbb{E}\left\|\boldsymbol{m}_n \odot \nabla f_n(\boldsymbol{w}_n^j(t), b_n^j(t))\right\|^2$$

$$\leq \eta^2(t)(i-1)^2 G^2, \tag{26}$$

where the last inequality results from Assumption 3. By combining (25) and (26), we have

$$\mathbb{E}\left\|\frac{1}{N}\sum_{n=1}^{N}\left(\nabla f_n(\boldsymbol{w}_n^i(t)) - \nabla f_n(\boldsymbol{w}_g(t))\right)\right\|^2 \leq L^2\eta^2(t)(i-1)^2 G^2. \tag{27}$$

By combining (18) and (22)−(27), we have

$$\mathbb{E}F(\boldsymbol{w}_g(t+1)) \leq \mathbb{E}F(\boldsymbol{w}_g(t)) + \frac{L}{2}N\tau^2\eta^2(t)\sum_{n=1}^{N}\xi_n^2 + \eta(t)\tau\Psi\sum_{n=1}^{N}\left(d_{\boldsymbol{w}}\gamma_n - \sum_{l=1}^{d_{\boldsymbol{w}}}(\boldsymbol{k}_n)_l\right)$$

$$- \frac{\eta(t)\tau}{2}\mathbb{E}\left\|\nabla F(\boldsymbol{w}_g(t))\right\|^2 + L^2\eta^3(t)G^2\frac{\tau(\tau-1)(2\tau-1)}{12}$$

$$- \frac{\eta(t)}{2}\sum_{n=1}^{N}\sum_{i=1}^{\tau}\left(\frac{1}{N} - LN\tau\gamma_n^2\eta(t)\right)\left\|\nabla f_n(\boldsymbol{w}_n^i(t))\right\|^2. \tag{28}$$

Since $\eta(t) = \eta \leq \frac{1}{LN^2\tau}$, we have $-\frac{\eta(t)}{2}\sum_{n=1}^{N}\sum_{i=1}^{\tau}\left(\frac{1}{N} - LN\tau\gamma_n^2\eta(t)\right)\left\|\nabla f_n(\boldsymbol{w}_n^i(t))\right\|^2 \leq 0$. By rearranging the terms in (28), we obtain

$$\mathbb{E}\left\|\nabla F(\boldsymbol{w}_g(t))\right\|^2 \leq \frac{2}{\eta\tau}\left(\mathbb{E}F(\boldsymbol{w}_g(t)) - \mathbb{E}F(\boldsymbol{w}_g(t+1))\right) + LN\tau\eta\sum_{n=1}^{N}\xi_n^2$$

$$+ 2\Psi\sum_{n=1}^{N}\left(d_{\boldsymbol{w}}\gamma_n - \sum_{l=1}^{d_{\boldsymbol{w}}}(\boldsymbol{k}_n)_l\right) + L^2\eta^2 G^2\frac{(\tau-1)(2\tau-1)}{6}. \tag{29}$$

Finally, we multiply both sides of (29) by $\frac{1}{T}$ and sum over $t = 1,\dots,T$. Then, Theorem 1 is concluded by considering that the first term on the right-hand side of (29) is a telescoping series. We have

$$\frac{2}{\eta\tau T}\sum_{t=0}^{T}\left(\mathbb{E}F(\boldsymbol{w}_g(t)) - \mathbb{E}F(\boldsymbol{w}_g(t+1))\right) = \frac{2}{\eta\tau T}\left(F(\boldsymbol{w}_g(1)) - \mathbb{E}F(\boldsymbol{w}_g(T+1))\right)$$

$$\leq \frac{2}{\eta\tau T}\left(F(\boldsymbol{w}_g(1)) - F^*\right), \tag{30}$$

where the last inequality is obtained by considering that $\mathbb{E}F(\boldsymbol{w}_g(t+1)) \geq F^*$.

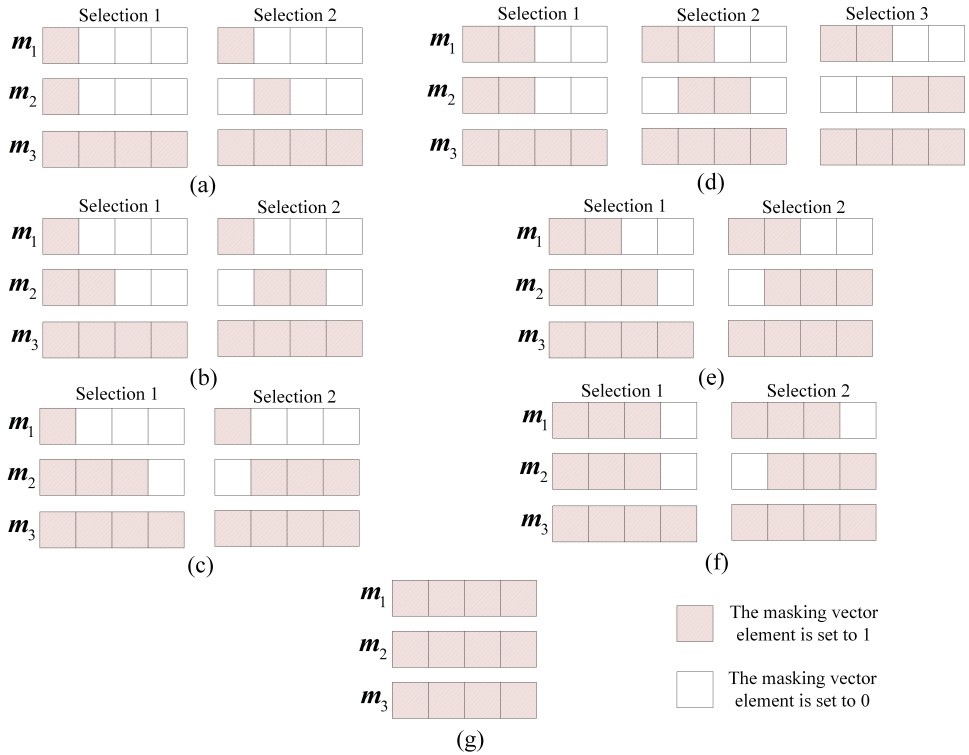

Figure 2: Illustration of masking vectors selection in a network with three devices. Device 3 has the maximum computational capability to train $\times 1$ parameters of the learning model. We consider different scenarios for computational capability of devices 1 and 2. (a) Devices 1 and 2 can train $\times 0.25$ parameters of the learning model. (b) Devices 1 and 2 can train $\times 0.25$ and $\times 0.5$ parameters of the learning model, respectively. (c) Devices 1 and 2 can train $\times 0.25$ and $\times 0.75$ parameters of the learning model, respectively. (d) Devices 1 and 2 can train $\times 0.5$ parameters of the learning model. (e) Devices 1 and 2 can train $\times 0.5$ and $\times 0.75$ parameters of the learning model, respectively. (f) Devices 1 and 2 can train $\times 0.75$ parameters of the learning model. (g) All three devices have the maximum computational capability.

## F    SELECTION OF MASKING VECTORS IN A TOY EXAMPLE

In a heterogeneous device setting, devices with lower computational capability partially train the learning model. Those devices use masking vectors to freeze a portion of the learning model during training based on their computational capability. As shown in Theorem 1, employing masking vectors lead to a bias term in the convergence bound. In this section, we demonstrate how the bias value is impacted by the design of masking vectors using a simple example. We consider three devices. One device has the maximum computational capability. However, different scenarios are considered for the computational capability of the other two devices. The devices aim to train a model with four parameters. Hence, their masking vectors have four elements. Fig. 2 shows the considered scenarios and the possible masking vectors selections, which may lead to different bias values.

For each scenario and for each possible selection, Table 6 shows vector $\boldsymbol{k}_n$ and the value of $\gamma_n$ for each device as well as the obtained bias value. For this example, the results in Table 6 illustrate that the bias value is minimized by freezing the same parameters for two devices with lower computational capability, in case each parameter cannot be trained at least by one of those devices (Figs. 2(a) and 2(b)). However, when each parameter can be trained at least by one of those less capable devices, the bias value is minimized by freezing distinct parameters for the devices (Figs. 2(c)−2(f)). These results are compatible with the empirical results obtained by Pfeiffer et al. (2022) and Yang et al. (2022), where random selection is used by the devices to prevent freezing of the same parameters.

Table 6: $\boldsymbol{k}_n$, $\gamma_n$, and the bias value for the possible masking vectors selections of the scenarios shown in Fig. 2. The selection with the minimum bias value is chosen based on Remark 1.

| Scenario | Possible selections | $\boldsymbol{k}_n$ | $\gamma_n$ | Bias value |
|---|---|---|---|---|
| Fig. 2(a) | **Selection 1** | $\boldsymbol{k}_1 = \boldsymbol{k}_2 = [\frac{1}{3},0,0,0]$ 
 $\boldsymbol{k}_3 = [\frac{1}{3},1,1,1]$ | $\gamma_1 = \gamma_2 = \frac{1}{3}$ 
 $\gamma_3 = 1$ | $\mathbf{\frac{8}{3}}$ |
| | Selection 2 | $\boldsymbol{k}_1 = [\frac{1}{2},0,0,0]$ 
 $\boldsymbol{k}_2 = [0,\frac{1}{2},0,0]$ 
 $\boldsymbol{k}_3 = [\frac{1}{2},\frac{1}{2},1,1]$ | $\gamma_1 = \frac{1}{2}$ 
 $\gamma_2 = \frac{1}{2}$ 
 $\gamma_3 = 1$ | 4 |
| Fig. 2(b) | **Selection 1** | $\boldsymbol{k}_1 = [\frac{1}{3},0,0,0]$ 
 $\boldsymbol{k}_2 = [\frac{1}{3},\frac{1}{2},0,0]$ 
 $\boldsymbol{k}_3 = [\frac{1}{3},\frac{1}{2},1,1]$ | $\gamma_1 = \frac{1}{3}$ 
 $\gamma_2 = \frac{1}{2}$ 
 $\gamma_3 = 1$ | $\mathbf{\frac{10}{3}}$ |
| | Selection 2 | $\boldsymbol{k}_1 = [\frac{1}{2},0,0,0]$ 
 $\boldsymbol{k}_2 = [0,\frac{1}{2},\frac{1}{2},0]$ 
 $\boldsymbol{k}_3 = [\frac{1}{2},\frac{1}{2},\frac{1}{2},1]$ | $\gamma_1 = \frac{1}{2}$ 
 $\gamma_2 = \frac{1}{2}$ 
 $\gamma_3 = 1$ | 4 |
| Fig. 2(c) | Selection 1 | $\boldsymbol{k}_1 = [\frac{1}{3},0,0,0]$ 
 $\boldsymbol{k}_2 = [\frac{1}{3},\frac{1}{2},\frac{1}{2},0]$ 
 $\boldsymbol{k}_3 = [\frac{1}{3},\frac{1}{2},\frac{1}{2},1]$ | $\gamma_1 = \frac{1}{3}$ 
 $\gamma_2 = \frac{1}{2}$ 
 $\gamma_3 = 1$ | $\frac{10}{3}$ |
| | **Selection 2** | $\boldsymbol{k}_1 = [\frac{1}{2},0,0,0]$ 
 $\boldsymbol{k}_2 = [0,\frac{1}{2},\frac{1}{2},\frac{1}{2}]$ 
 $\boldsymbol{k}_3 = [\frac{1}{2},\frac{1}{2},\frac{1}{2},\frac{1}{2}]$ | $\gamma_1 = \frac{1}{2}$ 
 $\gamma_2 = \frac{1}{2}$ 
 $\gamma_3 = \frac{1}{2}$ | **2** |
| Fig. 2(d) | Selection 1 | $\boldsymbol{k}_1 = \boldsymbol{k}_2 = [\frac{1}{3},\frac{1}{3},0,0]$ 
 $\boldsymbol{k}_3 = [\frac{1}{3},\frac{1}{3},1,1]$ | $\gamma_1 = \gamma_2 = \frac{1}{3}$ 
 $\gamma_3 = 1$ | $\frac{8}{3}$ |
| | Selection 2 | $\boldsymbol{k}_1 = [\frac{1}{2},\frac{1}{3},0,0]$ 
 $\boldsymbol{k}_2 = [0,\frac{1}{3},\frac{1}{2},0]$ 
 $\boldsymbol{k}_3 = [\frac{1}{2},\frac{1}{3},\frac{1}{2},1]$ | $\gamma_1 = \frac{1}{2}$ 
 $\gamma_2 = \frac{1}{2}$ 
 $\gamma_3 = 1$ | 4 |
| | **Selection 3** | $\boldsymbol{k}_1 = [\frac{1}{2},\frac{1}{2},0,0]$ 
 $\boldsymbol{k}_2 = [0,0,\frac{1}{2},\frac{1}{2}]$ 
 $\boldsymbol{k}_3 = [\frac{1}{2},\frac{1}{2},\frac{1}{2},\frac{1}{2}]$ | $\gamma_1 = \frac{1}{2}$ 
 $\gamma_2 = \frac{1}{2}$ 
 $\gamma_3 = \frac{1}{2}$ | **2** |
| Fig. 2(e) | Selection 1 | $\boldsymbol{k}_1 = [\frac{1}{3},\frac{1}{3},0,0]$ 
 $\boldsymbol{k}_2 = [\frac{1}{3},\frac{1}{3},\frac{1}{2},0]$ 
 $\boldsymbol{k}_3 = [\frac{1}{3},\frac{1}{3},\frac{1}{2},1]$ | $\gamma_1 = \frac{1}{3}$ 
 $\gamma_2 = \frac{1}{2}$ 
 $\gamma_3 = 1$ | $\frac{10}{3}$ |
| | **Selection 2** | $\boldsymbol{k}_1 = [\frac{1}{2},\frac{1}{3},0,0]$ 
 $\boldsymbol{k}_2 = [0,\frac{1}{3},\frac{1}{2},\frac{1}{2}]$ 
 $\boldsymbol{k}_3 = [\frac{1}{2},\frac{1}{3},\frac{1}{2},\frac{1}{2}]$ | $\gamma_1 = \frac{1}{2}$ 
 $\gamma_2 = \frac{1}{2}$ 
 $\gamma_3 = \frac{1}{2}$ | **2** |
| Fig. 2(f) | Selection 1 | $\boldsymbol{k}_1 = \boldsymbol{k}_2 = [\frac{1}{3},\frac{1}{3},\frac{1}{3},0]$ 
 $\boldsymbol{k}_3 = [\frac{1}{3},\frac{1}{3},\frac{1}{3},1]$ | $\gamma_1 = \gamma_2 = \frac{1}{3}$ 
 $\gamma_3 = 1$ | $\frac{8}{3}$ |
| | **Selection 2** | $\boldsymbol{k}_1 = [\frac{1}{2},\frac{1}{3},\frac{1}{3},0]$ 
 $\boldsymbol{k}_2 = [0,\frac{1}{3},\frac{1}{3},\frac{1}{2}]$ 
 $\boldsymbol{k}_3 = [\frac{1}{2},\frac{1}{3},\frac{1}{3},\frac{1}{2}]$ | $\gamma_1 = \frac{1}{2}$ 
 $\gamma_2 = \frac{1}{2}$ 
 $\gamma_3 = \frac{1}{2}$ | **2** |
| Fig. 2(g) | | $\boldsymbol{k}_1 = \boldsymbol{k}_2 = \boldsymbol{k}_3 = [\frac{1}{3},\frac{1}{3},\frac{1}{3},\frac{1}{3}]$ | $\gamma_1 = \gamma_2 = \gamma_3 = \frac{1}{3}$ | **0** |

The results in Table 6 also show that by increasing the computational capability of the devices, the bias value can be decreased. In the extreme case that all three devices have the maximum computational capability (i.e., Fig. 2(g)), the bias value is zero.

## G    EFFECT OF MASKING RATE ON THE COMPUTATIONS

By increasing the masking rate (i.e., $1 - \frac{\psi_n}{d_w}$), each device can reduce the number of trainable parameters and the number of FLOPs based on its computational capability. Fig. 3 shows the reduction percentage[7], which can be obtained for the number of trainable parameters and the number of FLOPs versus the masking rate. For the results in Fig. 3, we have considered that a device performs each local update iteration on a batch containing 50 data samples of CIFAR-10 and CIFAR-100 datasets using ResNet (PreResNet18) and MobileNet, respectively.

---

[7]Reduction percentage is the percentage change in a value compared to its maximum value.

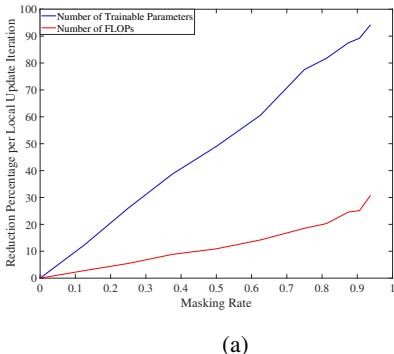
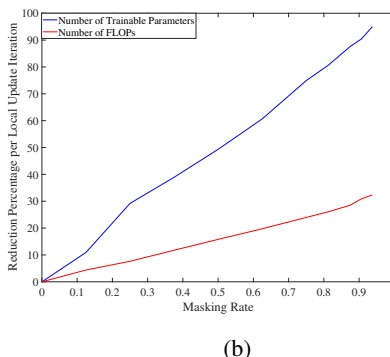

(a)

(b)

Figure 3: Reduction percentage for the number of trainable parameters and number of FLOPs versus masking rate for (a) ResNet (PreResNet18) and (b) MobileNet.

## H   SOLVING PROBLEM $\mathcal{P}^{\mathrm{MASK}}$ BY SCA

In this section, we first transform the non-convex constraints in optimization problem $\mathcal{P}^{\mathrm{mask}}$ into convex functions or a difference of two convex functions. Then, by using successive convex approximation (SCA) method, we can obtain a suboptimal solution for problem $\mathcal{P}^{\mathrm{mask}}$ in polynomial time. We first rewrite constraint (9a) in the form of the following inequalities:

$$(\tilde{\boldsymbol{k}}_n)_j \leq \frac{(\tilde{\boldsymbol{m}}_n)_j}{\sum_{n'=1}^{N} (\tilde{\boldsymbol{m}}_{n'})_j}, \ j \in \Lambda, \ n \in [N], \tag{31a}$$

$$(\tilde{\boldsymbol{k}}_n)_j \geq \frac{(\tilde{\boldsymbol{m}}_n)_j}{\sum_{n'=1}^{N} (\tilde{\boldsymbol{m}}_{n'})_j}, \ j \in \Lambda, \ n \in [N]. \tag{31b}$$

Since $(\tilde{\boldsymbol{m}}_n)_j$ is a binary variable, the non-convex constraint (31b) can be expressed as the following convex constraint:

$$(\tilde{\boldsymbol{k}}_n)_j \geq \frac{(\tilde{\boldsymbol{m}}_n)_j^2}{\sum_{n'=1}^{N} (\tilde{\boldsymbol{m}}_{n'})_j}, \ j \in \Lambda, \ n \in [N]. \tag{32}$$

The non-convex constraint (31a) can be rewritten as follows:

$$(\tilde{\boldsymbol{k}}_n)_j \sum_{n'=1}^{N} (\tilde{\boldsymbol{m}}_{n'})_j \leq (\tilde{\boldsymbol{m}}_n)_j, \ j \in \Lambda, \ n \in [N]. \tag{33}$$

Equality $(\tilde{\boldsymbol{k}}_n)_j \sum_{n'=1}^{N} (\tilde{\boldsymbol{m}}_{n'})_j = \frac{1}{2}\left( \left((\tilde{\boldsymbol{k}}_n)_j + \sum_{n'=1}^{N} (\tilde{\boldsymbol{m}}_{n'})_j\right)^2 - (\tilde{\boldsymbol{k}}_n)_j^2 - (\sum_{n'=1}^{N} (\tilde{\boldsymbol{m}}_{n'})_j)^2 \right)$ can be used to express the left-hand side of (33) as a difference of two convex functions. We have

$$\frac{1}{2}\left( \left((\tilde{\boldsymbol{k}}_n)_j + \sum_{n'=1}^{N} (\tilde{\boldsymbol{m}}_{n'})_j\right)^2 - (\tilde{\boldsymbol{k}}_n)_j^2 - (\sum_{n'=1}^{N} (\tilde{\boldsymbol{m}}_{n'})_j)^2 \right) \leq (\tilde{\boldsymbol{m}}_n)_j, \ j \in \Lambda, \ n \in [N]. \tag{34}$$

Next, we relax the binary constraint (9c) in the form of the difference of two convex functions as follows:

$$\sum_{n=1}^{N} \sum_{j \in \Lambda} (\tilde{\boldsymbol{m}}_n)_j - \sum_{n=1}^{N} \sum_{j \in \Lambda} (\tilde{\boldsymbol{m}}_n)_j^2 \leq 0, \tag{35a}$$

$$0 \leq (\tilde{\boldsymbol{m}}_n)_j \leq 1, \ j \in \Lambda, \ n \in [N]. \tag{35b}$$

Finally, we define the following functions:

$$\Theta((\tilde{\boldsymbol{m}}_n)_j) = (\tilde{\boldsymbol{m}}_n)_j^2, \tag{36a}$$

$$\vartheta((\tilde{\boldsymbol{k}}_n)_j, \tilde{\boldsymbol{M}}_j) = (\tilde{\boldsymbol{k}}_n)_j^2 + (\sum_{n'=1}^{N} (\tilde{\boldsymbol{m}}_{n'})_j)^2, \tag{36b}$$

where vector $\tilde{\boldsymbol{M}}_j = ((\tilde{\boldsymbol{m}}_{n'})_j, \ n' \in [N])$.

Algorithm 3 describes the SCA algorithm for solving problem $\mathcal{P}^{\text{mask}}$. Let $i$ denote the iteration index. In Line 1, we initialize the maximum number of iterations $i^{\text{max}}$. In Line 2, the decision variables $\tilde{\boldsymbol{m}}_n^{(1)}$ and $\tilde{\boldsymbol{k}}_n^{(1)}$, $n \in [N]$ are initialized in iteration $i = 1$ with a feasible solution of problem $\mathcal{P}^{\text{mask}}$. In Line 4, the optimal solution of problem $\mathcal{P}^{\text{mask-SCA}}$ (i.e., $\tilde{\boldsymbol{m}}_n^*$ and $\tilde{\boldsymbol{k}}_n^*$, $n \in [N]$) are determined. In Line 5, using $\tilde{\boldsymbol{m}}_n^*$ and $\tilde{\boldsymbol{k}}_n^*$, we update $\tilde{\boldsymbol{m}}_n^{(i+1)}$ and $\tilde{\boldsymbol{k}}_n^{(i+1)}$, $n \in [N]$ to be used for obtaining the first-order approximations $\hat{\Theta}((\tilde{\boldsymbol{m}}_n)_j)$ and $\hat{\vartheta}((\tilde{\boldsymbol{k}}_n)_j, \tilde{\boldsymbol{M}}_j)$ of functions $\Theta((\tilde{\boldsymbol{m}}_n)_j)$ and $\vartheta((\tilde{\boldsymbol{k}}_n)_j, \tilde{\boldsymbol{M}}_j)$, respectively. The iteration index is updated in Line 6. The steps within Lines 3 to 7 are repeated until the algorithm converges to a solution or we reach to $i^{\text{max}}$. The convex problem $\mathcal{P}^{\text{mask-SCA}}$, which is solved in each iteration $i$, is as follows:

$$\mathcal{P}^{\text{mask-SCA}} : \quad \underset{\tilde{\boldsymbol{m}}_n, \tilde{\boldsymbol{k}}_n, \epsilon_n, n \in [N]}{\text{minimize}} \quad \sum_{n=1}^{N} \left( d_{\boldsymbol{w}} \max_j (\tilde{\boldsymbol{k}}_n)_j - \sum_{j' \in \Lambda} |\pi_{j'}| (\tilde{\boldsymbol{k}}_n)_{j'} + \epsilon_n \right)$$

subject to constraints (9b), (32), (35b), and (9d),

$$\frac{1}{2} \left( \left( (\tilde{\boldsymbol{k}}_n)_j + \sum_{n'=1}^{N} (\tilde{\boldsymbol{m}}_{n'})_j \right)^2 - \hat{\vartheta}((\tilde{\boldsymbol{k}}_n^{(i)})_j, \tilde{\boldsymbol{M}}_j^{(i)}) \right) \le (\tilde{\boldsymbol{m}}_n)_j, \ j \in \Lambda, \ n \in [N],$$

$$\sum_{n=1}^{N} \sum_{j \in \Lambda} (\tilde{\boldsymbol{m}}_n)_j - \sum_{n=1}^{N} \sum_{j \in \Lambda} \hat{\Theta}((\tilde{\boldsymbol{m}}_n^{(i)})_j) \le 0.$$

---

**Algorithm 3:** SCA Algorithm for Solving Problem $\mathcal{P}^{\text{mask}}$

1: Set $i := 1$ and initialize the maximum number of iterations $i^{\text{max}}$.
2:     Initialize variables $\tilde{\boldsymbol{m}}_n^{(1)}$ and $\tilde{\boldsymbol{k}}_n^{(1)}$, $n \in [N]$.
3:     **Repeat**
4:        Determine the optimal solution $\tilde{\boldsymbol{m}}_n^*$ and $\tilde{\boldsymbol{k}}_n^*$, $n \in [N]$ of problem $\mathcal{P}^{\text{mask-SCA}}$.
5:        Update variables $\tilde{\boldsymbol{m}}_n^{(i+1)} := \tilde{\boldsymbol{m}}_n^*$ and $\tilde{\boldsymbol{k}}_n^{(i+1)} := \tilde{\boldsymbol{k}}_n^*$, $n \in [N]$.
6:        Set $i := i + 1$.
7:     **Until** $i = i^{\text{max}}$ or $\tilde{\boldsymbol{m}}_n$ and $\tilde{\boldsymbol{k}}_n$, $n \in [N]$ converge.
8:     **Return** $\tilde{\boldsymbol{m}}_n^{\text{opt}} := \tilde{\boldsymbol{m}}_n^{(i)}$ and $\tilde{\boldsymbol{k}}_n^{\text{opt}} := \tilde{\boldsymbol{k}}_n^{(i)}$, $n \in [N]$.

---

# I    PERFORMANCE COMPARISON FOR DOMAINNET DATASET

Different from the considered class non-IID configuration for CIFAR-10 and CIFAR-100 datasets, in this section, we evaluate the performance of our proposed algorithm under feature non-IID configuration. We perform our experiment with AlexNet on DomainNet dataset. The dataset contains images of six distinct domains including Clipart, Infograph, Painting, Quickdraw, Real, and Sketch. We consider 30 devices in the network, and split each domain among 5 devices. We set the number of communication rounds and the learning rate to be 100 and 0.01, respectively. Table 7 shows the obtained test accuracy after fine-tuning and the number of trainable parameters for PerFedMask compared to other baseline algorithms. The results in Table 7 indicate that PerFedMask can achieve a higher test accuracy compared to HeteroFL and Split-Mix FL algorithms, while the number of trainable parameters is much less than FedBABU, FedProx, FedNova, and FedAvg algorithms.

Table 7: Test accuracy after fine-tuning and number of trainable parameters of PerFed-Mask and the baseline algorithms for DomainNet dataset

| Test accuracy after fine-tuning | | | | | | |
|---|---|---|---|---|---|---|
| PerFedMask (Ours) | FedBABU | FedProx | FedNova | HeteroFL | Split-Mix FL | FedAvg |
| 70.68 | 72.29 | 71.95 | 72.85 | 67.68 | 68.44 | 72.24 |
| **Number of trainable parameters** | | | | | | |
| PerFedMask (Ours) | FedBABU | FedProx | FedNova | HeteroFL | Split-Mix FL | FedAvg |
| 31.221M | 57.022M | 57.063M | 57.063M | 28.983M | 4.058M | 57.063M |

## J    COMPARISON OF TRAINING CURVES

This section includes some of the training curves for our experiments. Fig. 4 illustrates the training loss over communication rounds for PerFedMask and the baseline algorithms on CIFAR-100 and DomainNet datasets. In Fig. 4, we have considered the full device participation scenario. The difference between the training loss of PerFedMask and FedBABU is the bias resulted by the device heterogeneity. Although we have minimized this bias through solving $\mathcal{P}^{\text{mask}}$, it has not been completely eliminated. Fine-tuning after training helps to improve the performance of PerFedMask. When $c = 0.1$, Fig. 5 shows the evolution of the validation accuracy over communication rounds for PerFedMask, FedBABU, FedAvg, and Split-Mix FL algorithms on CIFAR-10 and CIFAR-100 datasets.

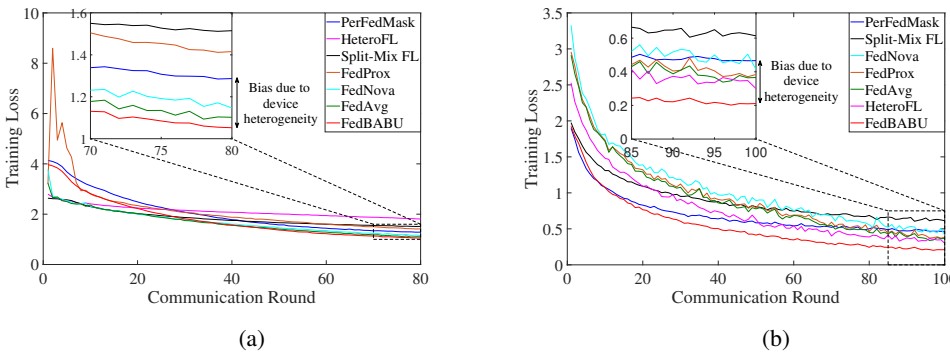

Figure 4: Training loss evolution over communication rounds for (a) CIFAR-100 and (b) DomainNet datasets. $c$ is set to 1.

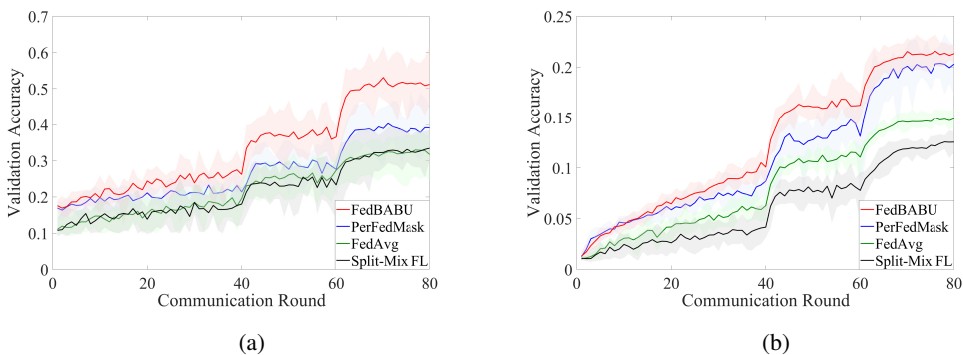

Figure 5: Validation accuracy evolution over communication rounds for (a) CIFAR-10 and (b) CIFAR-100 datasets. $c$ is set to 0.1.

## K    PERFORMANCE COMPARISON USING BOX PLOTS

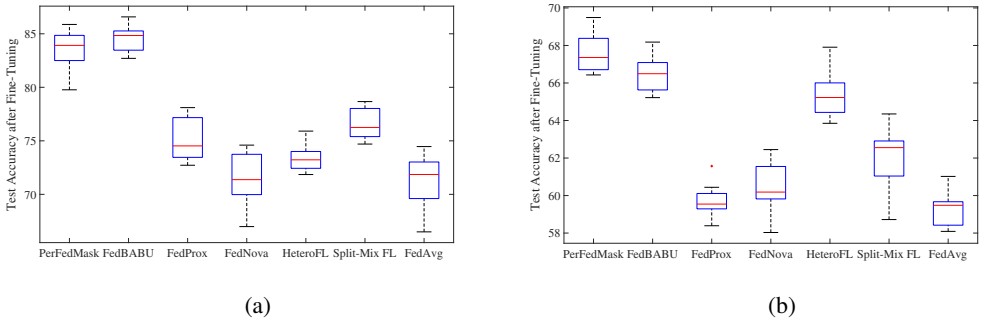

Figure 6: Test accuracy after fine-tuning for (a) CIFAR-10 and (b) CIFAR-100 datasets. $c$ is set to 0.1.

## L  COMBINATION OF PERFEDMASK WITH HETEROFL AND SPLIT-MIX FL

Table 8: Performance comparison on CIFAR-100 dataset when $c = 1$

| Algorithm | Test accuracy | | # of trainable parameters | # of FLOPs | |
|---|---|---|---|---|---|
| | Before fine-tuning | After fine-tuning | | Forward | Backward |
| PerFedMask + Split-Mix FL | 29.66 | 70.78 | 0.204M | 17.182M | 0.122G |
| PerFedMask + HeteroFL | 30.94 | 71.56 | 1.611M | 0.122G | 0.464G |
| PerFedMask | 32.04 | 72.40 | 1.803M | 0.220G | 0.711G |
| Split-Mix FL | 23.09 | 65.95 | 0.223M | 17.182M | 0.127G |
| HeteroFL | 30.13 | 68.65 | 1.774M | 0.122G | 0.473G |
| FedBABU | 29.70 | 69.01 | 3.207M | 0.220G | 0.863G |

## M  EFFECT OF FREEZING THE LOCAL HEAD MODELS DURING TRAINING

In this section, we investigate the impact of freezing the local head models on the test accuracy of PerFedMask. In particular, we compare the test accuracy of PerFedMask with (w/) and without (w/o) freezing the local head models. In the w/o head freezing scenario, the learning model is not decoupled into the global model and the local head model. Table 9 shows that the test accuracy of PerFedMask is improved by keeping the local head models frozen during training.

Table 9: Test accuracy of PerFedMask with and without freezing the local head models

| PerFedMask | CIFAR-10 dataset | | CIFAR-100 dataset | |
|---|---|---|---|---|
| | $c = 0.1$ | $c = 1$ | $c = 0.1$ | $c = 1$ |
| w/ head freezing | 83.60 | 88.43 | 67.47 | 72.40 |
| w/o head freezing | 71.56 | 85.76 | 54.85 | 67.89 |

## N  PROOF OF LEMMA 2

We define two auxiliary sequences $\bar{\boldsymbol{g}}(t) = \sum_{n=1}^{N} \boldsymbol{k}_n \odot \sum_{i=1}^{\tau} \nabla f_n(\boldsymbol{w}_n^i(t))$ and $\boldsymbol{g}(t) = \sum_{n=1}^{N} \boldsymbol{k}_n \odot \sum_{i=1}^{\tau} \nabla f_n(\boldsymbol{w}_n^i(t), b_n^i(t))$. We have

$$
\begin{aligned}
\mathbb{E}\left\|\boldsymbol{w}_g(t+1) - \boldsymbol{w}^*\right\|^2 &= \mathbb{E}\left\|\sum_{n=1}^{N} \boldsymbol{k}_n \odot \boldsymbol{w}_n^{\tau+1}(t) - \boldsymbol{w}^*\right\|^2 \\
&= \mathbb{E}\left\|\sum_{n=1}^{N} \boldsymbol{k}_n \odot \boldsymbol{w}_g(t) - \eta(t)\sum_{n=1}^{N} \boldsymbol{k}_n \odot \sum_{i=1}^{\tau} \nabla f_n(\boldsymbol{w}_n^i(t), b_n^i(t)) - \boldsymbol{w}^*\right\|^2 \\
&= \mathbb{E}\left\|\boldsymbol{w}_g(t) - \eta(t)\boldsymbol{g}(t) - \boldsymbol{w}^*\right\|^2 \\
&= \mathbb{E}\left\|\boldsymbol{w}_g(t) - \eta(t)\boldsymbol{g}(t) - \boldsymbol{w}^* + \eta(t)\bar{\boldsymbol{g}}(t) - \eta(t)\bar{\boldsymbol{g}}(t)\right\|^2 \\
&= \mathbb{E}\underbrace{\left\|\boldsymbol{w}_g(t) - \boldsymbol{w}^* - \eta(t)\bar{\boldsymbol{g}}(t)\right\|^2}_{A_1} + \eta^2(t)\mathbb{E}\underbrace{\left\|\boldsymbol{g}(t) - \bar{\boldsymbol{g}}(t)\right\|^2}_{A_2} \\
&\quad - 2\eta(t)\mathbb{E}\underbrace{\left\langle \boldsymbol{w}_g(t) - \boldsymbol{w}^* - \eta(t)\bar{\boldsymbol{g}}(t), \boldsymbol{g}(t) - \bar{\boldsymbol{g}}(t)\right\rangle}_{A_3}.
\end{aligned}
\tag{37}
$$

Since $\mathbb{E}\boldsymbol{g}(t) = \bar{\boldsymbol{g}}(t)$, we have $\mathbb{E}A_3 = 0$. Next, we focus on bounding $A_1$.

$$
A_1 = \left\|\boldsymbol{w}_g(t) - \boldsymbol{w}^* - \eta(t)\bar{\boldsymbol{g}}(t)\right\|^2 = \left\|\boldsymbol{w}_g(t) - \boldsymbol{w}^*\right\|^2 + \eta^2(t)\underbrace{\left\|\bar{\boldsymbol{g}}(t)\right\|^2}_{B_1} \underbrace{-2\eta(t)\left\langle \boldsymbol{w}_g(t) - \boldsymbol{w}^*, \bar{\boldsymbol{g}}(t)\right\rangle}_{B_2}.
\tag{38}
$$

By considering the convexity of $\|\cdot\|^2$ and by using Assumption 1, we can bound $B_1$ as follows:

$$
\begin{aligned}
B_1 = \|\bar{\boldsymbol{g}}(t)\|^2 &= \left\|\sum_{n=1}^{N}\sum_{i=1}^{\tau} \boldsymbol{k}_n \odot \nabla f_n(\boldsymbol{w}_n^i(t))\right\|^2 \\
&\leq N\tau \sum_{n=1}^{N}\sum_{i=1}^{\tau} \left\|\boldsymbol{k}_n \odot \nabla f_n(\boldsymbol{w}_n^i(t))\right\|^2 \\
&\leq N\tau \sum_{n=1}^{N}\sum_{i=1}^{\tau} \gamma_n^2 \left\|\nabla f_n(\boldsymbol{w}_n^i(t))\right\|^2 \\
&\leq 2LN\tau \sum_{n=1}^{N}\sum_{i=1}^{\tau} \gamma_n^2 \left(f_n(\boldsymbol{w}_n^i(t)) - f_n^*\right).
\end{aligned}
\tag{39}
$$

Next, we aim to bound $B_2$. We have

$$
\begin{aligned}
B_2 = &- 2\eta(t) \left\langle \boldsymbol{w}_g(t) - \boldsymbol{w}^*, \bar{\boldsymbol{g}}(t)\right\rangle \\
= &- 2\eta(t) \left\langle \boldsymbol{w}_g(t) - \boldsymbol{w}^*, \sum_{n=1}^{N}\sum_{i=1}^{\tau} \boldsymbol{k}_n \odot \nabla f_n(\boldsymbol{w}_n^i(t))\right\rangle \\
= &- 2\eta(t) \sum_{n=1}^{N}\sum_{i=1}^{\tau} \left\langle \boldsymbol{w}_g(t) - \boldsymbol{w}_n^i(t) + \boldsymbol{w}_n^i(t) - \boldsymbol{w}^*, \boldsymbol{k}_n \odot \nabla f_n(\boldsymbol{w}_n^i(t))\right\rangle \\
= &\ \eta(t) \sum_{n=1}^{N}\sum_{i=1}^{\tau} \underbrace{(-2)\left\langle \boldsymbol{w}_g(t) - \boldsymbol{w}_n^i(t), \boldsymbol{k}_n \odot \nabla f_n(\boldsymbol{w}_n^i(t))\right\rangle}_{C_1} \\
&+ 2\eta(t) \sum_{n=1}^{N}\sum_{i=1}^{\tau} \underbrace{(-1)\left\langle \boldsymbol{w}_n^i(t) - \boldsymbol{w}^*, \boldsymbol{k}_n \odot \nabla f_n(\boldsymbol{w}_n^i(t))\right\rangle}_{C_2}.
\end{aligned}
\tag{40}
$$

We first obtain an upper bound of $C_1$. We have

$$
\begin{aligned}
C_1 &\leq 2\left|\left\langle \boldsymbol{w}_g(t) - \boldsymbol{w}_n^i(t), \boldsymbol{k}_n \odot \nabla f_n(\boldsymbol{w}_n^i(t))\right\rangle\right| \\
&\overset{(a)}{\leq} 2\gamma_n \left\|\boldsymbol{w}_g(t) - \boldsymbol{w}_n^i(t)\right\| \left\|\nabla f_n(\boldsymbol{w}_n^i(t))\right\| \\
&\overset{(b)}{\leq} \frac{\gamma_n}{\eta(t)} \left\|\boldsymbol{w}_g(t) - \boldsymbol{w}_n^i(t)\right\|^2 + \eta(t)\gamma_n \left\|\nabla f_n(\boldsymbol{w}_n^i(t))\right\|^2 \\
&\overset{(c)}{\leq} \frac{\gamma_n}{\eta(t)} \left\|\boldsymbol{w}_g(t) - \boldsymbol{w}_n^i(t)\right\|^2 + 2L\eta(t)\gamma_n \left(f_n(\boldsymbol{w}_n^i(t)) - f_n^*\right),
\end{aligned}
\tag{41}
$$

where inequality (a) results from triangle and Hölder's inequalities. Inequality (b) results from the inequality of arithmetic and geometric means (AM-GM) inequality. For inequality (c), we use Assumption 1.

Next, we obtain an upper bound of $C_2$ by using Lemma 1. In particular, by considering $\boldsymbol{x} = \boldsymbol{w}^* - \boldsymbol{w}_n^i(t)$, $\boldsymbol{y} = \boldsymbol{k}_n$, and $\boldsymbol{z} = \nabla f_n(\boldsymbol{w}_n^i(t))$ in Lemma 1, we can bound $C_2$ as follows:

$$
\begin{aligned}
C_2 &\leq \gamma_n \left\langle \boldsymbol{w}^* - \boldsymbol{w}_n^i(t), \nabla f_n(\boldsymbol{w}_n^i(t))\right\rangle + \Omega\left(d_{\boldsymbol{w}}\gamma_n - \sum_{l=1}^{d_{\boldsymbol{w}}} (\boldsymbol{k}_n)_l\right) \\
&\leq -\gamma_n \left(f_n(\boldsymbol{w}_n^i(t)) - f_n(\boldsymbol{w}^*)\right) + \frac{\mu\gamma_n}{2} \underbrace{\left(-\left\|\boldsymbol{w}_n^i(t) - \boldsymbol{w}^*\right\|^2\right)}_{D} + \Omega\left(d_{\boldsymbol{w}}\gamma_n - \sum_{l=1}^{d_{\boldsymbol{w}}} (\boldsymbol{k}_n)_l\right),
\end{aligned}
\tag{42}
$$

where the last inequality results from Assumption 4. Now, we focus on bounding $D$. Considering the inequality $\|\boldsymbol{x} + \boldsymbol{y}\|^2 \leq 2\|\boldsymbol{x}\|^2 + 2\|\boldsymbol{y}\|^2$ for any $\boldsymbol{x}, \boldsymbol{y} \in \mathbb{R}^d$, and by replacing $\boldsymbol{x}$ with $\boldsymbol{w}_n^i(t) - \boldsymbol{w}^*$

and $\boldsymbol{y}$ with $\boldsymbol{w}_g(t) - \boldsymbol{w}_n^i(t)$ we have

$$D \leq \left\| \boldsymbol{w}_g(t) - \boldsymbol{w}_n^i(t) \right\|^2 - \frac{1}{2} \left\| \boldsymbol{w}_g(t) - \boldsymbol{w}^* \right\|^2. \tag{43}$$

By combining (38)−(43), we obtain

$$\begin{aligned}
\mathbb{E}A_1 \leq & \left( 1 - \frac{\mu\eta(t)\tau}{2} \sum_{n=1}^{N} \gamma_n \right) \mathbb{E}\|\boldsymbol{w}_g(t) - \boldsymbol{w}^*\|^2 \\
& + (1 + \mu\eta(t)) \sum_{n=1}^{N} \sum_{i=1}^{\tau} \gamma_n \, \mathbb{E}\left\| \boldsymbol{w}_g(t) - \boldsymbol{w}_n^i(t) \right\|^2 \\
& + 2L\eta^2(t) \sum_{n=1}^{N} \sum_{i=1}^{\tau} \left( N\tau\gamma_n^2 + \gamma_n \right) \mathbb{E}\left( f_n(\boldsymbol{w}_n^i(t)) - f_n^* \right) \\
& - 2\eta(t) \sum_{n=1}^{N} \sum_{i=1}^{\tau} \gamma_n \, \mathbb{E}\left( f_n(\boldsymbol{w}_n^i(t)) - f_n(\boldsymbol{w}^*) \right) \\
& + 2\eta(t)\tau\Omega \sum_{n=1}^{N} \left( d_{\boldsymbol{w}}\gamma_n - \sum_{l=1}^{d_{\boldsymbol{w}}} (\boldsymbol{k}_n)_l \right). \tag{44}
\end{aligned}$$

By rearranging the terms in (44), we have

$$\begin{aligned}
\mathbb{E}A_1 \leq & \left( 1 - \frac{\mu\eta(t)\tau}{2} \sum_{n=1}^{N} \gamma_n \right) \mathbb{E}\|\boldsymbol{w}_g(t) - \boldsymbol{w}^*\|^2 \\
& + (1 + \mu\eta(t)) \sum_{n=1}^{N} \sum_{i=1}^{\tau} \gamma_n \, \mathbb{E}\left\| \boldsymbol{w}_g(t) - \boldsymbol{w}_n^i(t) \right\|^2 \\
& - 2\eta(t) \sum_{n=1}^{N} \sum_{i=1}^{\tau} \gamma_n \left( 1 - L\eta(t) \left( N\tau\gamma_n + 1 \right) \right) \mathbb{E}\left( f_n(\boldsymbol{w}_n^i(t)) - f_n(\boldsymbol{w}^*) \right) \\
& + 2L\tau\eta^2(t) \sum_{n=1}^{N} \left( N\tau\gamma_n^2 + \gamma_n \right) \mathbb{E}(f_n(\boldsymbol{w}^*) - f_n^*) \\
& + 2\eta(t)\tau\Omega \sum_{n=1}^{N} \left( d_{\boldsymbol{w}}\gamma_n - \sum_{l=1}^{d_{\boldsymbol{w}}} (\boldsymbol{k}_n)_l \right). \tag{45}
\end{aligned}$$

Now, we aim to bound $f_n(\boldsymbol{w}_n^i(t)) - f_n(\boldsymbol{w}^*)$ as follows:

$$\begin{aligned}
f_n(\boldsymbol{w}_n^i(t)) - f_n(\boldsymbol{w}^*) &= \left( f_n(\boldsymbol{w}_n^i(t)) - f_n(\boldsymbol{w}_g(t)) \right) + \left( f_n(\boldsymbol{w}_g(t)) - f_n(\boldsymbol{w}^*) \right) \\
&\overset{(a)}{\geq} \left\langle \nabla f_n(\boldsymbol{w}_g(t)), \boldsymbol{w}_n^i(t) - \boldsymbol{w}_g(t) \right\rangle + \left( f_n(\boldsymbol{w}_g(t)) - f_n(\boldsymbol{w}^*) \right) \\
&\overset{(b)}{\geq} -\frac{\eta(t)}{2} \left\| \nabla f_n(\boldsymbol{w}_g(t)) \right\|^2 - \frac{1}{2\eta(t)} \left\| \boldsymbol{w}_g(t) - \boldsymbol{w}_n^i(t) \right\|^2 \\
&\quad + \left( f_n(\boldsymbol{w}_g(t)) - f_n(\boldsymbol{w}^*) \right) \\
&\overset{(c)}{\geq} -L\eta(t) \left( f_n(\boldsymbol{w}_g(t)) - f_n^* \right) - \frac{1}{2\eta(t)} \left\| \boldsymbol{w}_g(t) - \boldsymbol{w}_n^i(t) \right\|^2 \\
&\quad + \left( f_n(\boldsymbol{w}_g(t)) - f_n(\boldsymbol{w}^*) \right), \tag{46}
\end{aligned}$$

where inequality (a) results from the convexity of $f_n(\boldsymbol{w})$, inequality (b) is obtained by using Cauchy-Schwarz and AM-GM inequalities, and inequality (c) is due to Assumption 1.

By combining (26), (45), and (46), we have

$$
\begin{aligned}
\mathbb{E} A_1 \leq & \left(1 - \frac{\mu \eta(t) \tau}{2} \sum_{n=1}^{N} \gamma_n \right) \mathbb{E}\|\boldsymbol{w}_g(t) - \boldsymbol{w}^*\|^2 \\
& + \eta^2(t) G^2 \frac{\tau(\tau-1)(2\tau-1)}{6} \sum_{n=1}^{N} \gamma_n \left(2 + \mu \eta(t) - L\eta(t)\left(N\tau\gamma_n + 1\right)\right) \\
& - 2\eta(t)\tau\left(1 - L\eta(t)\right) \sum_{n=1}^{N} \gamma_n \left(1 - L\eta(t)\left(N\tau\gamma_n + 1\right)\right) \mathbb{E}(f_n(\boldsymbol{w}_g(t)) - f_n(\boldsymbol{w}^*)) \\
& + 2L\tau\eta^2(t) \sum_{n=1}^{N} \gamma_n \left(1 + \left(1 - L\eta(t)\right)\left(N\tau\gamma_n + 1\right)\right) \mathbb{E}(f_n(\boldsymbol{w}^*) - f_n^*) \\
& + 2\eta(t)\tau\Omega \sum_{n=1}^{N} \left(d_{\boldsymbol{w}} \gamma_n - \sum_{l=1}^{d_{\boldsymbol{w}}} (\boldsymbol{k}_n)_l \right). \quad (47)
\end{aligned}
$$

**Lemma 3.** *For* $\eta(t) \leq \frac{1}{L(N\tau+1)}$, *for all* $t = 1, \ldots, T$, *we have*

$$
\begin{aligned}
& - 2\eta(t)\tau\left(1 - L\eta(t)\right) \sum_{n=1}^{N} \gamma_n \left(1 - L\eta(t)\left(N\tau\gamma_n + 1\right)\right) \mathbb{E}(f_n(\boldsymbol{w}_g(t)) - f_n(\boldsymbol{w}^*)) \\
& \leq 2\eta(t)\tau\left(1 - L\eta(t)\right) \Upsilon \sum_{n=1}^{N} \left(\gamma_n - \frac{1}{N}\right) - L\eta(t)\left(\gamma_n\left(N\tau\gamma_n + 1\right) - \frac{1}{N}\left(\tau + 1\right)\right). \quad (48)
\end{aligned}
$$

*Proof.* See Appendix P. $\qquad \square$

Using Lemma 3, we can simplify (47) as follows:

$$
\begin{aligned}
\mathbb{E} A_1 \leq & \left(1 - \frac{1}{2}\mu\tau \sum_{n=1}^{N} \gamma_n\, \eta(t) \right) \mathbb{E}\|\boldsymbol{w}_g(t) - \boldsymbol{w}^*\|^2 \\
& + \eta^2(t) G^2 \frac{\tau(\tau-1)(2\tau-1)}{6}(2 + \mu) \sum_{n=1}^{N} \gamma_n \\
& + 2\eta(t)\tau\Upsilon \sum_{n=1}^{N} \left(\gamma_n - \frac{1}{N}\right) \\
& + 2L\tau\eta^2(t) \sum_{n=1}^{N} \gamma_n\left(2 + N\tau\gamma_n\right) \Gamma_n \\
& + 2\eta(t)\tau\Omega \sum_{n=1}^{N} \left(d_{\boldsymbol{w}} \gamma_n - \sum_{l=1}^{d_{\boldsymbol{w}}} (\boldsymbol{k}_n)_l \right). \quad (49)
\end{aligned}
$$

Finally, we use (20) to bound $\mathbb{E} A_2$ in (37). Lemma 2 is concluded by combining (20), (37), and (49).

## O  PROOF OF THEOREM 2

First, through induction, we show that for a diminishing stepsize $\eta(t) = \frac{2}{q_0(t+\kappa)}$, we have $\delta(t) \leq \frac{\beta(t)}{t+\kappa}$, where $\beta(t) = \max\{\frac{2q_1}{q_0}(t+\kappa) + \frac{4q_2}{q_0^2}, (\kappa+1)\delta(1)\}$. Note that the considered $\eta(t)$ satisfies the mentioned condition in Lemma 2. Moreover, the definition of $\beta(t)$ ensures that the inequality

holds for $\delta(1)$. Now, we assume that the inequality holds for $t$. We show that it also holds for $t+1$. From Lemma 2, we have

$$\delta(t+1) \leq (1 - q_0\,\eta(t))\,\delta(t) + q_1\,\eta(t) + q_2\,\eta^2(t)$$

$$\leq \left(1 - \frac{2}{t+\kappa}\right)\frac{\beta(t)}{t+\kappa} + \frac{2q_1}{q_0\,(t+\kappa)} + \frac{4q_2}{q_0^2\,(t+\kappa)^2}$$

$$= \frac{t+\kappa-1}{(t+\kappa)^2}\beta(t) + \frac{2q_0q_1\,(t+\kappa) + 4q_2}{q_0^2\,(t+\kappa)^2} - \frac{\beta(t)}{(t+\kappa)^2}$$

$$\leq \frac{t+\kappa-1}{(t+\kappa)^2}\beta(t)$$

$$\leq \frac{t+\kappa-1}{(t+\kappa)^2 - 1}\beta(t)$$

$$\leq \frac{\beta(t)}{t+\kappa+1}. \tag{50}$$

Finally, by the $L$-smoothness assumption for $F$, we have

$$\mathbb{E}F(\boldsymbol{w}_g(T)) - F^* \leq \frac{L}{2}\delta(T) \leq \frac{L}{2}\frac{\beta(T)}{T+\kappa}. \tag{51}$$

From the definition of $\beta(t)$, we have $\beta(T) \leq \frac{2q_1}{q_0}\,(T+\kappa) + \frac{4q_2}{q_0^2} + (\kappa+1)\,\delta(1)$. Combining this with (51) completes the proof of Theorem 2.

## P    PROOF OF LEMMA 3

First, we define two vectors $\boldsymbol{x}$ and $\boldsymbol{y} \in \mathbb{R}^N$, where $(\boldsymbol{x})_n = \gamma_n\,(1 - L\eta(t)\,(N\tau\gamma_n + 1))$ and $(\boldsymbol{y})_n = f_n(\boldsymbol{w}_g(t)) - f_n(\boldsymbol{w}^*)$, $n \in [N]$, respectively. Let $\boldsymbol{X}$ and $\boldsymbol{Y}$, respectively, denote the corresponding diagonal matrices of vectors $\boldsymbol{x}$ and $\boldsymbol{y}$. By using Theorem 3 in Fang et al. (1994), we have the following inequality:

$$\text{Tr}(\boldsymbol{X}\boldsymbol{Y}) \geq \lambda_N(\boldsymbol{X})\,\text{Tr}(\boldsymbol{Y}) + \lambda_N(\boldsymbol{Y})\,(\text{Tr}(\boldsymbol{X}) - N\lambda_N(\boldsymbol{X})), \tag{52}$$

where $\lambda_N(\boldsymbol{X})$ and $\lambda_N(\boldsymbol{Y})$ are the smallest eigenvalue of matrices $\boldsymbol{X}$ and $\boldsymbol{Y}$, respectively. Since $\boldsymbol{X}$ and $\boldsymbol{Y}$ are diagonal matrices, we have $\lambda_N(\boldsymbol{X}) = \min_{n\in[N]}\,(\boldsymbol{x})_n$ and $\lambda_N(\boldsymbol{Y}) = \min_{n\in[N]}\,(\boldsymbol{y})_n$. By considering $\eta(t) \leq \frac{1}{L(N\tau+1)}$, all the elements of vector $\boldsymbol{x}$ including $\lambda_N(\boldsymbol{X})$ are nonnegative. Also, $(\boldsymbol{x})_n$ is a quadratic function with respect to $\gamma_n$. For $\frac{1}{N} \leq \gamma_n \leq 1$, by considering that $\eta(t) \leq \frac{1}{L(N\tau+1)}$, it can be verified that the minimum value of $(\boldsymbol{x})_n$ is obtained at $\gamma_n = \frac{1}{N}$. Thus, $\lambda_N(\boldsymbol{X})$ is lower bounded as $\lambda_N(\boldsymbol{X}) \geq \frac{1}{N}\,(1 - L\eta(t)\,(\tau+1))$. Hence, we have

$$-2\eta(t)\tau\,(1 - L\eta(t))\sum_{n=1}^{N}\gamma_n\,(1 - L\eta(t)\,(N\tau\gamma_n + 1))\,(f_n(\boldsymbol{w}_g(t)) - f_n(\boldsymbol{w}^*))$$

$$\overset{(a)}{\leq} -2\eta(t)\tau\,(1 - L\eta(t))\,\lambda_N(\boldsymbol{X})\sum_{n=1}^{N}f_n(\boldsymbol{w}_g(t)) - f_n(\boldsymbol{w}^*)$$

$$\quad -2\eta(t)\tau\,(1 - L\eta(t))\min_{n\in[N]}\,(f_n(\boldsymbol{w}_g(t)) - f_n(\boldsymbol{w}^*))\,(\text{Tr}(\boldsymbol{X}) - N\lambda_N(\boldsymbol{X}))$$

$$\overset{(b)}{\leq} -2\eta(t)\tau\,(1 - L\eta(t))\,\lambda_N(\boldsymbol{X})N\,(F(\boldsymbol{w}_g(t)) - F^*)$$

$$\quad +2\eta(t)\tau\,(1 - L\eta(t))\,\Upsilon\,(\text{Tr}(\boldsymbol{X}) - N\lambda_N(\boldsymbol{X}))$$

$$\overset{(c)}{\leq} 2\eta(t)\tau\,(1 - L\eta(t))\,\Upsilon\left(\sum_{n=1}^{N}\gamma_n\,(1 - L\eta(t)\,(N\tau\gamma_n + 1)) - (1 - L\eta(t)\,(\tau+1))\right), \tag{53}$$

where for inequality (a), we use (52). We also use the fact that $-2\eta(t)\tau\,(1 - L\eta(t)) \leq 0$. Note that since $\eta(t) \leq \frac{1}{L(N\tau+1)}$, $\forall t$, the quadratic function $-2\eta(t)\tau\,(1 - L\eta(t))$ is always nonpositive. Inequality (b) results from the definition of $F(\boldsymbol{w})$. For inequality (c), since $F(\boldsymbol{w}_g(t)) \geq F^*$, we use the fact that $-2\eta(t)\tau\,(1 - L\eta(t))\,\lambda_N(\boldsymbol{X})N\,(F(\boldsymbol{w}_g(t)) - F^*) \leq 0$. Finally, Lemma 3 is concluded by using the obtained lower bound for $\lambda_N(\boldsymbol{X})$ and by rearranging the terms in (53).

