# OpenReview forum: "PerFedMask: Personalized Federated Learning with Optimized Masking Vectors"
_ICLR.cc/2023/Conference — ICLR 2023 poster_

### Official Review · Reviewer_pTHn · 2022-10-24

**Confidence:** 3
**Correctness:** 3
**Technical Novelty And Significance:** 2
**Empirical Novelty And Significance:** 2
**Recommendation:** 6

**Clarity, Quality, Novelty And Reproducibility:**

The paper is mostly clear and relatively straightforward to follow. One thing that could be improved in the exposition is Eq. 1; it is not entirely correct for FL, as this objective can be solved without the devices communicating altogether since the optimization problems are independent. FL seeks consensus between the local models, which is what correlates the optimization problems and creates the global model. I would advise the authors to update it. Other than that, I believe that there is reasonable information to be able to reproduce some of the results, although I would have liked some more information on the optimization specifics for the SCA algorithm implementation (e.g., an algorithm, hyperparameters it have, etc).

In my opinion, the main drawback of this work is novelty. As far as I understand, the personalization component is just an application of FedBABU whereas the freezing part comes from the likes of works such as CoCo-FL. The novelty of this work thus seems to lie on the convergence analysis which leads to the uncovering of the convergence bound bias and the server side algorithm to fix it. Having said that, there is no comparison against the heuristic strategy proposed at CoCo-FL so it is hard to tell its significance compared to prior art.

Finally, some more questions from my side:
- It is a bit weird that in Table 3, more finetuning steps can be harmful. Is it due to overfitting on the local dataset that this happens?
- What is the importance of freezing the local heads on each client? How does the method perform without the local head freezing?


**Strength And Weaknesses:**

Strengths
- Extensive experimental evaluation
- Theoretical analysis of convergence

Weaknesses:
- Server side optimization problem for masking does not consider FLOPs directly (which is a better proxy for computational cost, e.g., convolutional layers are more computationally expensive that linear ones, however the latter have more parameters)
- Incremental; mostly a combination of FedBABU + freezing methods (such as CoCo-FL)
- No indication of uncertainty / error bars in the results (so unclear whether the gains in some cases are significant)

**Summary Of The Paper:**

This work proposes a specific masking strategy on each client that leads to two outcomes; 1) adhering to the computational constraints of each client and 2) improving the personalised model performance. Instead of prior works that “zero out” the masked parameters, this work instead “freezes” the masked parameters to the values communicated by the server. While strictly speaking this leads to less efficiency gains, the benefit is that each hardware constrained client can maintain the flexibility of the server model, while still being able to contribute in training (a part of) it. To optimise the masking assignment, the authors formalise it as a constraint satisfaction problem that targets to minimise the theoretical convergence bound bias coming from masking. As the latter is an NP-hard problem, the authors propose a specific approximation procedure that seems to empirically work. Finally, the improvement to the personalised model performance comes from also adopting the FedBABU strategy, i.e., splitting the learning of the body and head of the network; the head is frozen and kept local while training the (masked) global body and when the body has converged, the entire network is fine-tuned locally with the available data samples. The authors perform extensive experiments and ablation studies to show the performance of PerFedMask in various settings.

**Summary Of The Review:**

Based on all of the above, and primarily due to my novelty concerns, I cannot recommend acceptance. This work could benefit by better positioning in terms of works such as CoCo-FL and by making more clear the improvements that it brings to the table.

---

> ### Author Response · Authors · 2022-11-15
> **Response to Reviewer pTHn - Part 1**
>
> We would like to thank the reviewer for his/her comments that have helped us to improve our work. We respond to your questions and address your comments below.
>
> **Q1.** Server side optimization problem for masking does not consider FLOPs directly.
>
> **Authors' Response:**  We have provided Fig. 3 in Appendix G to show how the maximum number of trainable parameters can be selected by a device based on its computational capability, i.e, the number of FLOPs that can be performed by a device at each local update iteration. The curves in Appendix G have been obtained by averaging over possible masking configurations. Thus, by using the maximum number of parameters that can be trained by each device in the server side optimization problem, the effect of the number of FLOPs has been indirectly considered in our formulation. We agree with the reviewer that it would be better to directly consider the number of FLOPs in the server side optimization problem. However, the objective function in the optimization problem is based on the number of parameters. Also, the masking vectors, which correspond to the learning parameters, are decision variables of the formulated optimization problem. Using the maximum number of trainable parameters in the optimization problem constraints therefore allows solving the optimization problem effectively with the aid of SCA algorithm and the available optimization solvers. If we use the number of FLOPs (or other metrics such as time as mentioned by the second reviewer), heuristic algorithms are required to solve the optimization problem. This is due to the fact that there is not any equation that demonstrates the connection between the number of parameters and the number of FLOPs. This work is our first attempt to optimize the selection of masking vectors by minimizing the convergence bound. Unfortunately, due to the time limit, we could not implement and provide the results for replacing the number of parameters with the number of FLOPs. In the conclusion section, we have pointed it out that considering freezing priority for different layers in the neural network architecture is a method that we consider as future work. We believe that considering freezing priority based on the number of FLOPs for each layer and the effect of each layer on the final accuracy can further improve the results.
>
> **Q2.** Incremental; mostly a combination of FedBABU + freezing methods (such as CoCo-FL).
>
> **Authors' Response:** (1) As we stated in related works, using masking vectors to address the device heterogeneity issue itself is not completely novel in both freezing or pruning methods. However, in our work, different from the previous works, we have provided strong theoretical motivation for systematically designing the masking vectors via an optimization framework. (2) Since the bias cannot be eliminated completely by minimizing the bias term in the convergence bound, we used the idea of decoupling the learning model into a global model and a local head model, and performing fine-tuning after convergence to a global model. It is true that the idea comes from the FedBABU work, which only considers the data heterogeneity issue in federated learning. However, we have utilized this idea for another purpose: improving the performance of federated learning in a heterogeneous device setting. (3) CoCoFL is an unpublished arXiv work, which is concurrent to our work. The key differences between our work and CoCoFL work are as follows: First, in CoCoFL, a heuristic configuration selection is proposed. In particular, each device has a set of configurations. For each configuration, some of the layers of the neural network architecture are considered to be frozen. Then, each device randomly selects one of the configurations based on its available resources in each communication round. However, in our proposed algorithm, PerFedMask, the masking vectors are selected by solving an optimization problem before training. Second, although in CoCoFL, the authors provide some results for non-IID data settings, there is not any specific approach in their proposed algorithm to address the data heterogeneity issue. However, in PerFedMask, by using the decoupling and fine-tuning approaches, we could kill two birds with one stone, i.e., addressing both the data and device heterogeneity issues in federated learning.
>
> **Q3.** No indication of uncertainty / error bars in the results (so unclear whether the gains in some cases are significant).
>
> **Authors' Response:** We thank the reviewer for this comment.  Following the reviewer's comment, we have included Fig. 6 in Appendix K of the revised paper to show more statistics of the results.

---

> ### Author Response · Authors · 2022-11-15
> **Response to Reviewer pTHn - Part 2**
>
> **Q4.** One thing that could be improved in the exposition is Eq. 1; it is not entirely correct for FL, as this objective can be solved without the devices communicating altogether since the optimization problems are independent. FL seeks consensus between the local models, which is what correlates the optimization problems and creates the global model. I would advise the authors to update it.
>
> **Authors' Response:**  As suggested by the reviewer, we have updated the corresponding paragraph and equation in the revised paper. In the previous version, we followed the common formulation used in the personalized federated learning studies such as Collins et al., (2021). In particular, there was an emphasis on the personalized model for each device in the previous formulation. In addition, it is true that each device can optimize its local objective function based on its local data samples. However, due to the limited number of local data samples at the devices, they will collaborate with each other to obtain better learning models through consensus.
>
> **Q5.** There is reasonable information to be able to reproduce some of the results, although I would have liked some more information on the optimization specifics for the SCA algorithm implementation (e.g., an algorithm, hyperparameters it have, etc).
>
> **Authors' Response:** We have included more details about the SCA method including its algorithm and hyperparameters in Appendix H.
>
> **Q6.** The main drawback of this work is novelty. As far as I understand, the personalization component is just an application of FedBABU whereas the freezing part comes from the likes of works such as CoCo-FL. The novelty of this work thus seems to lie on the convergence analysis which leads to the uncovering of the convergence bound bias and the server side algorithm to fix it. Having said that, there is no comparison against the heuristic strategy proposed at CoCo-FL so it is hard to tell its significance compared to prior art.
>
> **Authors' Response:** In FedBABU work, the authors empirically showed that it is critically important to train all the layers in the global model during local update iterations to prevent performance degradation (Appendix F3 in Oh et al. (2022)). In our work, we showed that we can use masking vectors to freeze some parts of the learning model for each device, but still obtain the same or better performance than FedBABU. We believe that this improvement can be taken into account. In particular, instead of masking the same layers of the global model for all the devices, in PerFedMask, different devices mask different layers of the global model to minimize the bias term. The trained layers are then aggregated at the server using Eq. (3) in the revised paper. Thus, compared to FedBABU, the performance can be improved in our proposed algorithm. Regarding the work CoCoFL, we have properly cited this work in the related works. In Table 5, we have shown the effectiveness of our proposed algorithm compared to the random masking selection. Please note that CoCoFL is an unpublished arXiv work, which is concurrent to our work. When we, as a reviewer, checked the ICLR 2023 Reviewer Guide, we noticed that the paper should not be judged through comparison with the very recent "published" paper (on or after May 28, 2022).
>
> **Q7.** It is a bit weird that in Table 3, more fine-tuning steps can be harmful. Is it due to overfitting on the local dataset that this happens?
>
> **Authors' Response:** Yes, we believe that the reason of this variation is due to overfitting of the training data samples. We noticed that for fine-tuning, the learning rate and the number of fine-tuning steps should be selected carefully.
>
> **Q8.** What is the importance of freezing the local heads on each client? How does the method perform without the local head freezing?
>
> **Authors' Response:** Freezing the local head models has two benefits. First, it provides a flexibility to the learning model to be better fine-tuned using the local data samples at each device. Second, it accelerates fine-tuning to the local data samples. Specifically, instead of starting fine-tuning from the head model obtained through federated learning, fine-tuning is started from the randomly initialized local head model. As suggested by the reviewer, we have included the following table in Appendix N to compare the performance of PerFedMask with and without local head freezing. The results clearly show the importance of freezing the local head models.
>
> | PerFedMask | CIFAR-10, $c=0.1$| CIFAR-10, $c=1$| CIFAR-100, $c=0.1$|CIFAR-100, $c=1$|
> |----------- |----------- |----------- |----------- |----------- |
> with head freezing|83.60 |88.43 |67.47 |72.40|
> without head freezing|71.56 |85.76 |54.85| 67.89|

---

> ### Author Response · Authors · 2022-11-15
> **Response to Reviewer pTHn - Part 3**
>
> **Q9.** Based on all of the above, and primarily due to my novelty concerns, I cannot recommend acceptance. This work could benefit by better positioning in terms of works such as CoCo-FL and by making more clear the improvements that it brings to the table.
>
> **Authors' Response:** Our contributions are as follows: (1) We theoretically showed that using masking vectors to address the device heterogeneity issue leads to a bias term in the convergence bound. (2) We systematically designed the masking vectors by minimizing the bias term at the server before training. (3) We empirically showed that the decoupling and fine-tuning idea, which was previously proposed to address data heterogeneity issue, can also improve federated learning performance in the heterogeneous device settings. (4) We compared the performance of our proposed algorithm in terms of accuracy, number of trainable parameters, and number of FLOPs with the state-of-the-art FL algorithms in the literature.
>
> We would like to thank the reviewer for his/her valuable comments. We hope that we have addressed the comments in a satisfactory manner.

---

> ### Comment · Reviewer_pTHn · 2022-11-18
> **Response to rebuttal**
>
> I would like to thank the authors for addressing some of my concerns. I will raise my score to a 6, as I still believe that this work is a bit incremental given the context of related work.

---

> > ### Author Response · Authors · 2022-11-26
> > **Survey on Related Work**
> >
> > Dear Reviewer pTHn,
> >
> > We want to thank you for taking the time to review our updated paper and our replies to your comments. We really appreciate your valuable feedback. Based on your comments, we have conducted a literature survey on freezing method, where the learning model is partially trained on the devices with lower computational capability in federated learning. To the best of our knowledge, the related works include the following:
> >
> > * Sidahmed et al. (2021) proposed FedPT, where a portion of the learning model parameters are frozen during the entire training process for all the devices. Hence, in FedPT, a portion of the learning model parameters are untrained in order to reduce communication cost and computational complexity.
> >
> > * Chen et al. (2021) proposed APF to adaptively set the freezing periods for the learning model parameters. In particular, in APF, each stable parameter is frozen for a certain number of communication rounds and then unfrozen to determine if it requires more training. This approach requires an APF manager in the devices to identify the stabilized parameters and to control the freezing period for each parameter.
> >
> > * Cox et al. (2022) in [R1] proposed Aergia, which aims to speed up the training process for the devices with lower computational capability. In Aergia, the slow devices freeze the part of their model that is the most computationally intensive to train. Thus, those devices only train the unfrozen part of their model. However, each slow device offloads the training of its frozen part to another faster device. Aergia requires centralized scheduling. In particular, the server should identify the devices that would slow down the entire process. Additionally, the server should determine where those slow devices should offload the frozen part of their learning model for training. Also, the devices’ dataset similarities should be considered for offloading the training of the model from one slow device to another powerful device.
> >
> > * Yang et al. (2022) proposed PVT as a partial variable training method in federated learning. For PVT, the authors investigate three freezing approaches: (1) All devices freeze the same and fixed set of the learning model parameters in all communication rounds. (2) All devices freeze the same set of the learning model parameters which is selected randomly in each communication round. (3) Each device randomly selects its own frozen learning model parameters in each communication round. In PVT, the loss in accuracy is compensated through increasing the number of local update iterations and the number of devices participating in training in each communication round.
> >
> > * Pfeiffer et al. (2022) proposed CoCoFL, where a set of configurations is considered for each device. For each configuration, some of the layers of the neural network architecture are considered to be frozen. Then, each device randomly selects one of the configurations based on its available resources in each communication round. In CoCoFL, the authors also proposed to quantize the frozen parameters of the learning model during local update iterations to further speed up the computations for the devices with lower computational capability.
> >
> > We are the first to analytically investigate the impact of freezing parameters on the federated learning convergence bound. Different from the aforementioned related works, we proposed to systematically choose the frozen parameters for each device through solving an optimization problem before training. Our theoretical results show that in freezing methods, it is important to freeze distinct parameters for the devices in order that eventually, all the learning model parameters can be trained by at least one of the less capable devices. In particular, an example shown in Appendix F clarifies how PerFedMask systematically chooses which parts of the learning model to be trained by each device based on its computational capability.
> >
> > [R1] Bart Cox, Lydia Y Chen, and Jeremie Decouchant. Aergia: Leveraging heterogeneity in federated learning systems. In *Proc. of ACM/IFIP Int’l Middleware Conf.*, Quebec, Canada, Nov. 2022.

---

> ### Author Response · Authors · 2022-12-05
> **Providing Additional Clarification**
>
> Dear Reviewer pTHn,
>
> We appreciate your new evaluation. As the end of the discussion period is approaching, we would be happy to discuss if you still have any concerns. Further to the summary on the related work, we would like to provide the following clarifications:
>
> * **Theoretical contribution:**  The existing works on freezing method in the literature empirically show that using the masking vectors to partially train the learning model on devices with lower computational capability can degrade the final accuracy. Our theoretical analysis confirms this observation.
>
> * **Novelty over existing masking-based methods:** There are two major limitations of the alternative methods: (i) limited practical value with additional requirements in federated learning (FL) and (ii) lacking theoretical and performance guarantee. Our proposed method can address those two limitations. First, our method is **more practical** compared with the other methods, which require additional operations or modules. For example, Yang et al. (2022) require more local update iterations; Cox et al. (2022) require centralized scheduling for offloading; Chen et al. (2021) require additional manager modules. However, we follow the classical FL training setting without additional control modules or training steps. Our proposed method, PerFedMask, is more **flexible and practical** since it can be combined with other FL methods. Nevertheless, we have compared with the existing state-of-the-art FL algorithms and demonstrated PerFedMask's superior performance.  In addition, our method is the first to propose systematically designing the masking vectors via bias minimization, thus **guaranteeing model utility**. Concurrently with our work, CoCoFL, mentioned by the reviewer, uses the maximum number of parameters as an unfrozen part of the learning model for each device without ensuring accuracy improvement.
>
> * **Novelty over FedBABU:** Please note that the decoupling and fine-tuning ideas have been used in prior studies on long-tailed recognition (e.g., [R1]). However, Oh et al. (2022) in FedBABU have employed these ideas to address the data heterogeneity issue in FL. We used these ideas in PerFedMask to address the device heterogeneity issue in a heterogenous data setting in FL. Furthermore, compared with FedBABU, we proposed novel masking and optimization strategies, together with theoretical analysis. Leveraging an existing technique that was designed for a different purpose is not our limitation. Instead, it was our discovery that empowered our proposed masking strategy. Finally, because we are focusing on different problems, the experimental setup and evaluations are very different from those in FedBABU.
>
> * **Contribution to future work:** As mentioned by Reviewer Frgf, our method is “novel”, “principled”, and “motivated by the theoretic analysis of convergence”. We believe that the formulated optimization problem in our work can be generalized to consider other constraints, such as freezing priority for the learning model parameters. Furthermore, our proposed method can be exploited in other studies, where masking vectors are used to address the device heterogeneity issue such as pruning methods.
>
> [R1] Bingyi Kang, Saining Xie, Marcus Rohrbach, Zhicheng Yan, Albert Gordo, Jiashi Feng, and Yannis Kalantidis. Decoupling representation and classifier for long-tailed recognition. In *Proc. of Int’l Conf. on Learning Representations (ICLR)*, Apr. 2020.

---

### Official Review · Reviewer_nizq · 2022-10-25

**Confidence:** 4
**Correctness:** 3
**Technical Novelty And Significance:** 1
**Empirical Novelty And Significance:** 2
**Recommendation:** 3

**Clarity, Quality, Novelty And Reproducibility:**

The paper is easy to follow. The novelty is a little bit incremental given the existing work FedBABU.

**Strength And Weaknesses:**

Strengths:
1. The motivation to let devices with more computational resources to do more training makes sense.
2. There are empirical benefits of the proposed method regarding the accuracy and FLOPS.

Weaknesses/Questions:
1. How is \phi_g updated?
2. Can you show and compare the training curves?
3. Can you confirm that among those devices with lower computational capacity, a higher computational capacity leads to more trainable parameters?
4. In Eq (1), the \theta_n in f_n(\theta_n) is the concatenation of w and \phi. However, in section 4 the arguments becomes w, contradicting the previous definition. What are the definitions of F_n and f_n exactly? Why theorem1 is not affected by data heterogeneity?
5. In remark 1, minimizing upper bounds does not mean it is the best way.
6. There is a lacking in the empirical validation for the benefits (e.g., time) of the proposed method when applying to heterogeneous devices.


**Summary Of The Paper:**

The author proposes to replace the random mask in FedBABU with a masked optimized for the device heterogeneity. The proposed masking is layer-wise so that all parameters in a layer will be frozen or not, based on the formulation Eq(10). Empirical evaluation shows better performance, less trainable parameters and FLOPS.

**Summary Of The Review:**

The motivation of applying customized mask to FedBABU seems valid, but it is not surprising to see the empirical improvements. There are multiple aspects to further improve the paper as stated in weaknesses.

---

> ### Author Response · Authors · 2022-11-15
> **Response to Reviewer nizq - Part 1**
>
> We would like to thank the reviewer for his/her comments that have helped us to improve our work. We respond to your questions and address your comments below.
>
> **Q1.** How is $\phi_g$ updated?
>
> **Authors' Response:**  $\phi_g$ is never updated. As explained in the second paragraph of Section 3, for each device $n \in [N]$, the learning model $\theta_n$ is decoupled into $w_{n}$ and $\phi_n$. Before training, $\phi_g$ is initialized randomly, and for all the devices, $\phi_n$ is set to $\phi_g$ (Line 2 of Algorithm 1). During training, $\phi_n$ and $\phi_g$ are never updated by the devices or the server. In particular, to obtain the global model $w_{g}$, during training, each device $n$ only updates $w_{n}$ and sends it to the server for aggregation (Lines 5 to 12 of Algorithm 1). Thus, at the end of the communication rounds, for each device $n$, $\theta_n$ is the concatenation of the global model $w_{g}$ and $\phi_n=\phi_g$. Then, each device can obtain its personalized learning model including its specific local head model $\phi_n$ by fine-tuning (Lines 14-16 of Algorithm 1). In summary, we have used $\phi_g$ in Algorithm 1 to emphasize that the same randomly initialized $\phi_n$ is considered for all the devices $n \in [N]$. Regarding this question, we have updated the second paragraph of Section 3 in the revised paper.
>
> **Q2.** Can you show and compare the training curves?
>
> **Authors' Response:**  As suggested by the reviewer, we have included the training curves in Appendix J of the revised paper.
>
> **Q3.** Can you confirm that among those devices with lower computational capacity, a higher computational capacity leads to more trainable parameters?
>
> **Authors' Response:** Yes, floating-point operations (FLOPs) per second is a measure for computational capacity. For example, the Raspberry Pi 4 Model B is capable of about 13.5 billion FLOPs per second. As shown in Fig. 3 in Appendix G, a device with higher computational capability can choose lower masking rate to train on average more parameters per local update iteration. As illustrated in Fig. 3, lower masking rate also leads to less reduction on the number of FLOPs per local update iteration for devices with higher computational capabilities.
>
> **Q4.** In Eq (1), the $\theta_n$ in $f_n(\theta_n)$ is the concatenation of $w$ and $\phi$. However, in section 4 the arguments becomes $w$, contradicting the previous definition. What are the definitions of $F_n$ and $f_n$ exactly? Why theorem1 is not affected by data heterogeneity?
>
> **Authors' Response:** We thank the reviewer for pointing out this potential confusion. We have updated the second paragraph of Section 3 to make the notation more clear. The definitions of $F$ and $f_n$ have been mentioned in the second and first paragraphs of Section 3, respectively. $f_n$ is the expected loss over data samples of device $n$. Hence, $f_n$ is the local objective function for each device $n$. However, $F$ is the global objective function, which is obtained by averaging over all the devices' local objective functions. We do not have $F_n$ in any of our equations.
>
>
> The effect of data heterogeneity in Theorem 1 can be shown by the help of the last term in the right-hand side of inequality (28). Then, using the smoothness assumption for $f_n$, the effect of data heterogeneity can be extracted . In this paper, our main focus is to investigate the effect of device heterogeneity on the convergence bound. Hence, we tried to keep the proof of Theorem 1 simple by only investigating the effect of device heterogeneity on the convergence bound. Please note that due to the strong convexity assumption, less algebraic manipulations are required for showing the effect of data heterogeneity in Lemma 2 and Theorem 2. Hence, we have shown the effect of both data and device heterogeneities in the obtained convergence bound for strongly convex loss functions.

---

> ### Author Response · Authors · 2022-11-15
> **Response to Reviewer nizq - Part 2**
>
> **Q5.** In remark 1, minimizing upper bounds does not mean it is the best way.
>
> **Authors' Response:**  In federated learning, the devices aim to collaboratively find the global optimal point $w^{*}$  of the objective function $F$. Tighter convergence bound leads to a solution, which is closer to the global optimal point. This fact can be verified from (15) in Theorem 2. For example in a scenario that there is no device heterogeneity, $q_1$ is set to zero in (15). Using this upper bound, one can obtain the number of required communication rounds $T$ to reach the expected error smaller than $\epsilon$ for strongly convex functions. However, when there is device heterogeneity, $q_1$ is not equal to zero and due to the bias in the convergence bound, the expected error smaller than $\epsilon$ is not achievable only by increasing $T$. Thus, to obtain a tighter convergence bound, we have proposed to minimize the convergence bound by designing the masking vectors.
>
> To further elaborate on the significance of a tighter upper bound, we refer to some of the related works in the literature. The expected classification error is a highly non-convex function. Thus, it is a standard approach in supervised classification to minimize the log-loss function as an upper bound to the classification error. Roux (2017) in [R1] have shown that the log-loss function is not a tight upper bound after some initial iterations. Hence, it is proposed in [R1] to replace the log-loss function with a tighter upper bound for obtaining an improved classifier. Also, the idea of the tighter convergence bound has been used in several federated learning studies. Karimireddy et al. (2020) in [R2]  have obtained a tighter upper bound for FedAvg by using two separate step-sizes for local updates at the devices and aggregation at the server. Based on the obtained tighter convergence upper bound, it is shown in [R2] that the expected error smaller than $\epsilon$ can be reached by lower number of communication rounds. The authors in [R3]$-$[R5] have considered resource allocation among the devices  for supporting federated learning in resource-limited wireless networks. In all of these works, the authors have proposed their resource allocation schemes based on minimizing the convergence upper bounds.
>
> [R1] Nicolas Le Roux. Tighter bounds lead to improved classifiers. In *Proc. of Int’l Conf. on Learning Representations (ICLR)*, Toulon, France, Apr. 2017.
>
> [R2] Sai Praneeth Karimireddy, Satyen Kale, Mehryar Mohri, Sashank Reddi, Sebastian Stich, and Ananda Theertha Suresh. SCAFFOLD: Stochastic controlled averaging for federated learning. In *Proc. of Int'l Conf. on Machine Learning (ICML)*, Jul. 2020.
>
> [R3] Yuxuan Sun, Sheng Zhou, Zhisheng Niu, and Deniz Gunduz. Dynamic scheduling for over-the-air federated edge learning with energy constraints. *IEEE J. Sel. Areas Commun.*, 40(1):227–242, Jan. 2022.
>
> [R4] Zhibin Wang, Yong Zhou, Yuanming Shi, and Weihua Zhuang. Interference management for over-the-air federated learning in multi-cell wireless networks. *IEEE J. Sel. Areas Commun.*, 40(8):2361–2377, Jun. 2022.
>
> [R5] Wei Guo, Ran Li, Chuan Huang, Xiaoqi Qin, Kaiming Shen, and Wei Zhang. Joint device selection and power control for wireless federated learning. *IEEE J. Sel. Areas Commun.*, 40(8):2395–2410, Aug. 2022.
>
> **Q6.** There is a lacking in the empirical validation for the benefits (e.g., time) of the proposed method when applying to heterogeneous devices.
>
> **Authors' Response:**  We follow the common practice of previous studies related to federated learning in the heterogeneous device setting (e.g., Diao et al. (2021);  Sidahmed et al. (2021);  Hong et al. (2022)) to validate the benefits of our proposed method in terms of average number of trainable parameters and average number of FLOPs. Please note that FLOPs are unaffected by the implementation of low-level APIs, hardware, or the randomization of executions. However, due to the sensitivity of the wall time measurements to the execution environment, comparing the time may hurt the reproducibility of the results. As suggested by the reviewer, we have added the following table in Appendix L to compare the average processing time (in seconds) of the forward and backward propagation paths in our proposed algorithm with some of the baseline algorithms on CIFAR-10 dataset. In the following table, group 1 means devices with lower computational capability and group 2 means devices with the maximum computational capability:
> | | Forward time for group 1| Backward time for group 1| Forward time for group 2| Backward time for group 2|
> | -----------  | ----------- | ----------- | ----------- | ----------- |
> PerFedMask + Split-Mix FL| 0.91| 1.54| 8.51| 16.9|
> PerFedMask + HeteroFL|1.02| 1.66| 9.26| 29.05|
> PerFedMask|9.22 |11.51 |9.15 |27.68|
> Split-Mix FL|0.94| 2.11 |8.49 |17.11|
> HeteroFL|1.04 |2.54| 9.18 |29.12|
> FedBABU|9.19 |31.78 |9.21 |29.85|

---

> ### Author Response · Authors · 2022-11-15
> **Response to Reviewer nizq - Part 3**
>
> **Q7.** The novelty is a little bit incremental given the existing work FedBABU.
>
> **Authors' Response:**  There are main differences between our work and the existing work FedBABU: (1) Our main focus in this work is on the device heterogeneity issue in federated learning. However, FedBABU has been proposed as a personalized federated learning algorithm to only tackle the data heterogeneity issue. Thus, there is not any masking vector in FedBABU. (2) We have provided strong theoretical motivation for our proposed algorithm. In particular, we have theoretically shown that using masking vectors to tackle the device heterogeneity issue leads to a bias term in the convergence bound. (3) To prevent the performance degradation due to the bias, our proposed algorithm, PerFedMask, has two fundamental attributes. The first attribute is that the masking vectors are designed by minimizing the bias term. The second attribute is that the learning model is decoupled into a global model and a local head model, while fine-tuning is performed after convergence to a global model. (4) The first attribute is one of the contributions of our work. This attribute helps to alleviate the performance degradation caused by bias. However, the bias cannot be eliminated completely by this approach. (5) To further improve the performance of the federated learning with optimized masking vectors, we have incorporated the second attribute in PerFedMask. The idea of the second attribute comes from the FedBABU work. However, we have utilized this idea for another purpose: improving the performance of federated learning in a heterogeneous device setting, because the bias cannot be eliminated completely by the optimized masking vectors. (6) Since the second attribute has been proposed in FedBABU to address the data heterogeneity issue, by using this idea, we could kill two birds with one stone, i.e., addressing both the data and device heterogeneity issues in federated learning. Table 10 in Appendix N of the revised paper shows the significance of using the second attribute in PerFedMask.
>
> **Q8.** The motivation of applying customized mask to FedBABU seems valid, but it is not surprising to see the empirical improvements. There are multiple aspects to further improve the paper as stated in weaknesses.
>
> **Authors' Response:** In FedBABU work, the authors empirically showed that it is critically important to train all the layers in the global model during local update iterations to prevent performance degradation (Appendix F3 in Oh et al. (2022)). In our work, we showed that we can use masking vectors to freeze some parts of the learning model for each device, but still obtain the same or better performance than FedBABU. We believe that this improvement can be taken into account. In particular, instead of masking the same layers of the global model for all the devices, in PerFedMask, different devices mask different layers of the global model to minimize the bias term. The trained layers are then aggregated at the server using Eq. (3) in the revised paper. Thus, compared to FedBABU, the performance can be improved in our proposed algorithm.
>
> We would like to thank the reviewer for his/her valuable comments. We hope that we have addressed the comments in a satisfactory manner.

---

> ### Author Response · Authors · 2022-11-30
> **Rebuttal Follow-up**
>
> Dear Reviewer nizq,
>
> We would appreciate it greatly if you can re-evaluate our paper and check whether we have addressed your comments in a satisfactory manner. We would welcome the opportunity to discuss any remaining concerns and questions that you may have.

---

### Official Review · Reviewer_Frgf · 2022-10-25

**Confidence:** 4
**Correctness:** 4
**Technical Novelty And Significance:** 3
**Empirical Novelty And Significance:** 3
**Recommendation:** 8

**Clarity, Quality, Novelty And Reproducibility:**

Quality: The method has been compared to multiple baselines under two datasets. It would be better if the authors could provide more datasets.
Novelty: The proposed method is novel, and the insights are also novel through the lens of convergence.
Clarity: The writing could be improved by making the formulations more concise and by elaborating on theoretic results. Some notations should be more meaningful. For example, when *the normalized mask* (k_n)_l is introduced, the meaning of the term should be explained, in addition to its formulation.  Lemma 1 is not explained. Especially how it is used for proving Theorem 1? In Theorem 1, the meaning of each term should be explained more.

Minor issues with the clarity.
* The heads model should be head models.
* In the first sentence of the 2nd paragraph of Page 2, "may lead to a bias in the obtained solution of the learning model," is confusing. I suppose the bias should be on the convergence error rather than the model itself.

**Strength And Weaknesses:**

Strength
* The method is principled and motivated by the theoretic analysis of convergence. It is novel that the authors show the bias induced by heterogeneous masking. The finding reveals the critical upper bound of empirical studies of masked federated learning so far.
* Because of the general non-convex and masking-agnostic assumptions, the method can be integrated with different masked federated learning methods, like HeteroFL, Split-Mix FL, etc.
* The experiments also demonstrate that optimizing masks could improve different baselines. Therefore, the theorems and methods are generalizable and could be impactful in device-heterogeneous settings.

Weakness
* The method is motivated by the bias on the theoretic convergence bound. But the intuition behind the bias is not well-explained. The authors may elaborate on why the bias occurs. Instead of attributing the bias to the masking, it should be discussed how the bias is related to the heterogeneity. If all clients follow the same mask, how will the bias be? If all clients follow non-overlapped masks, will the bias be maximized? When the mask makes the bias minimized if can all clients afford the same number of parameters?
* Authors provided two techniques for solving (10a-d). More details about how to choose one of the two techniques should be provided to ease the practice.
* How does the method work on feature non-iid datasets? Since Split-Mix FL has shown that the HeteroFL would perform poorly on feature non-iid datasets, the Split-Mix FL based evaluation should be conducted on feature non-iid datasets.

**Summary Of The Paper:**

The work studied masked federated learning (FL) for device-heterogeneous settings. The theorems on the convergence of masked FL revealed the limitation of existing masking strategies that induced biases to the model convergence. Therefore, the authors propose to optimize the per-client masks on the server. Empirical results demonstrate that the proposed method can improve multiple existing masked federated learning methods.

**Summary Of The Review:**

The paper provides a novel insight into masked federated learning, which should be valuable for device-heterogeneous federated learning. Some issues on the clarity and experiments should be addressed during the rebuttal.

---

> ### Author Response · Authors · 2022-11-15
> **Response to Reviewer Frgf**
>
> We would like to thank the reviewer for his/her comments that have helped us to improve our work. We respond to your questions and address your comments below.
>
>
> **Q1.** The intuition behind the bias is not well-explained. The authors may elaborate on why the bias occurs. It should be discussed how the bias is related to the heterogeneity. If all clients follow the same mask, how will the bias be? If all clients follow non-overlapped masks, will the bias be maximized? When the mask makes the bias minimized if can all clients afford the same number of parameters?
>
> **Authors' Response:** To clarify how the bias is related to device heterogeneity, we have updated some parts in Section 4. In particular, devices with lower computational capability partially train the global model due to the zero-valued elements in their masking vectors. In Theorem 1, we show that employing the masking vectors to address the device heterogeneity issue in FL leads to a bias term in the convergence bound. In addition, Appendix F has been included in the revised paper to discuss the impact of the selection of masking vectors for the devices. In summary, when distinct parameters can be frozen by the devices and each parameter can be trained at least by one of the less capable devices, the bias is maximized by freezing the same parameters for the devices.
>
> **Q2.** Authors provided two techniques for solving (10a-d). More details about how to choose one of the two techniques should be provided to ease the practice.
>
> **Authors' Response:** We thank the reviewer for pointing out this potential confusion. We have updated Appendix H to clearly describe how the optimization problem in Section 5 can be solved by successive convex approximation (SCA) algorithm.
>
> **Q3.** How does the method work on feature non-iid datasets? Since Split-Mix FL has shown that the HeteroFL would perform poorly on feature non-iid datasets, the Split-Mix FL based evaluation should be conducted on feature non-iid datasets.
>
> **Authors' Response:** As suggested by the reviewer, we have included the results for DomainNet dataset in Appendix I as an experiment under feature non-IID configuration. The results in the following table show that our proposed algorithm, PerFedMask, outperforms both HeteroFL and Split-Mix FL algorithms in terms of test accuracy on DomainNet dataset.
> | |PerFedMask|FedBABU|FedProx|FedNova|HeteroFL|Split-Mix FL|FedAvg|
> | ----------- |----------- |----------- |----------- |----------- |----------- |----------- |----------- |
> Test accuracy|70.68|72.29| 71.95| 72.85| 67.68| 68.44 |72.24|
> Trainable parameters|31.221M |57.022M |57.063M |57.063M |28.983M |4.058M |57.063M|
>
> **Q4.** The meaning of the term $(k_n)_l$ should be explained, in addition to its formulation. Lemma 1 is not explained. Especially how it is used for proving Theorem 1? In Theorem 1, the meaning of each term should be explained more.
>
> **Authors' Response:** For each device $n$, vector $k_n$ is obtained as a normalized masking vector using the vectors $m_n$, $n \in \mathcal{N}(t)$. Using $k_n$ in (3) indicates that the server only aggregates the updated parameters from the participating devices. We have updated the revised paper based on the reviewer's suggestions. We also made changes related to the reviewer's comments for minor issues with the clarity.
>
> **Q5.** The paper provides a novel insight into masked federated learning, which should be valuable for device-heterogeneous federated learning. Some issues on the clarity and experiments should be addressed during the rebuttal.
>
> **Authors' Response:**  We have included more results in the revised paper. In particular, we have added Appendix I to compare the performance of PerFedMask with the baseline algorithms under feature non-IID configuration. In Appendix J, we have included training curves. Appendix K includes more statistics for some of the results presented in Table 1. Appendix N shows the effect of freezing the local head models on the performance of PerFedMask. To discuss more about how the design of the masking vectors affects the convergence, we have included Appendix F. Also, Appendix H has been updated to clearly describe the SCA algorithm.
>
> We would like to thank the reviewer for his/her valuable comments. We hope that we have addressed the comments in a satisfactory manner.

---

> ### Comment · Reviewer_Frgf · 2022-11-29
> **Acknowledge of the rebuttal**
>
> The authors have addressed most of my concerns, and I am raising my score.

---

> > ### Author Response · Authors · 2022-12-05
> > **Thanks**
> >
> > Dear Reviewer Frgf,
> >
> > We want to thank you for taking the time to review our updated paper and our replies to your comments. We appreciate the reviewer acknowledging the novelty and significance of our work.

---

### Author Response · Authors · 2022-11-15
**Thanks for your time and valuable feedback**

We would like to thank the reviewers for their constructive comments which helped us to improve our work. In the revised manuscript, we have used blue color for text that was changed or added based on the reviewers' comments. The responses to the reviewers' comments are given below.  We hope that we have addressed the comments in a satisfactory manner.

---

### Decision · Program_Chairs · 2023-01-20

**Decision:**

Accept: poster

**Justification For Why Not Higher Score:**

One reviewer was a marginal accept.

**Justification For Why Not Lower Score:**

One reviewer was a strong accept, and the other's premise for having a marginal accept is that the authors did not compare to/discuss a work that is actually concurrent with the submitted work.

**Metareview: Summary, Strengths And Weaknesses:**

The paper presents an extension of the personalized federated learning algorithm FedBABU that adds masking vectors associated with each device in order to mitigate device heterogeneity. The contribution over prior masking approaches to coping with device heterogeneity in the literature is that the authors show a bias term in the convergence bound introduced by the use of masking vectors and optimize the masking vectors to address this bias. Extensive experimental evaluation is provided supporting the performance of the method in different settings.

**Note From Pc:**

if the above contains the word "oral" or "spotlight" please see: "oral" presentation means -> notable-top-5% and "spotlight" means -> notable-top-25%. As stated in our emails, we are disassociating presentation type from AC recommendations

**Summary Of Ac-Reviewer Meeting:**

No meeting: the authors' comment to the AC about Reviewer nizq were taken into consideration. Namely, the novelty with respect to FedBABU is clear, and there is no need to provide wall-clock times in the evaluation of FL algorithms. Accordingly, I discarded this reviewer's score, which was the only reject.